# Speech and music recruit frequency-specific distributed and overlapping cortical networks

**Noémie te Rietmolen[1,2]\*, Manuel R Mercier[2], Agnès Trébuchon[1,2,3], Benjamin Morillon[1,2†], Daniele Schön[1,2\*†]**

[1]Institute for Language, Communication, and the Brain, Aix-Marseille University, Marseille, France; [2]Aix Marseille Université, INSERM, INS, Institut de Neurosciences des Systèmes, Marseille, France; [3]APHM, Hôpital de la Timone, Service de Neurophysiologie Clinique, Marseille, France

**Abstract** To what extent does speech and music processing rely on domain-specific and domain-general neural networks? Using whole-brain intracranial EEG recordings in 18 epilepsy patients listening to natural, continuous speech or music, we investigated the presence of frequency-specific and network-level brain activity. We combined it with a statistical approach in which a clear operational distinction is made between *shared*, *preferred*, and domain-*selective* neural responses. We show that the majority of focal and network-level neural activity is shared between speech and music processing. Our data also reveal an absence of anatomical regional selectivity. Instead, domain-selective neural responses are restricted to distributed and frequency-specific coherent oscillations, typical of spectral fingerprints. Our work highlights the importance of considering natural stimuli and brain dynamics in their full complexity to map cognitive and brain functions.

**\*For correspondence:**
noemieter@gmail.com (NtR);
daniele.schon@univ-amu.fr (DS)

†These authors contributed equally to this work

**Competing interest:** The authors declare that no competing interests exist.

## eLife assessment

This study presents **valuable** intracranial findings on how two types of natural auditory stimuli - speech and music - are processed in the human brain, and demonstrates that speech and music largely share network-level brain activities, thus challenging the domain-specific processing view. The evidence supporting the claims of the authors is **solid**. The work will be of broad interest to speech and music researchers as well as cognitive scientists in general.

## Introduction

The advent of neuroscience continues the longstanding debate on the origins of music and language—that fascinated Rousseau and Darwin (*Kivy, 1959*; *Rousseau, 2009*)—on new biological ground: evidence for the existence of selective and/or shared neural populations involved in their processing. The question on functional selectivity versus domain-general mechanisms is closely related to the question of the nature of the neural code: Are representations *sparse* (and localized) or *distributed*? While the former allows to explicitly represent any stimulus in a small number of neurons, it would require an intractable number of neurons to represent all possible stimuli. Experimental evidence instead suggests that stimulus identification is achieved through a population code, implemented by neural coupling in a distributed dynamical system (*Bizley and Cohen, 2013*; *Rissman and Wagner, 2012*). The question of the nature of the neural code has tremendous implications: it defines an epistemological posture on how to map cognitive and brain functions. This, in turn, affects both the

definition of cognitive operations—what is actually computed—as well as the way we look at the data—looking for differences or similarities.

Neuroimaging studies report mixed evidence of selectivity and resource sharing. On one hand, one can find claims for a clear distinction between brain regions exclusively dedicated to language versus other cognitive processes (*Chen et al., 2023*; *Fedorenko et al., 2011*; *Fedorenko and Blank, 2020*; *Friederici, 2020*) and for the existence of specific and separate neural populations for speech, music, and song (*Boebinger et al., 2021*; *Norman-Haignere et al., 2022*). On the other hand, other neuroimaging studies suggest that the brain regions that support language and speech also support nonlinguistic functions (*Albouy et al., 2020*; *Fadiga et al., 2009*; *Koelsch, 2011*; *Menon et al., 2002*; *Robert et al., 2023*; *Schön et al., 2010*). This point is often put forward when interpreting the positive impact music training can have on different levels of speech and language processing (*Flaugnacco et al., 2015*; *François et al., 2013*; *Kraus and Chandrasekaran, 2010*; *Schön et al., 2004*).

Several elements may account for these different findings. The very first may rely on the definition of a brain region. This can be considered as a set of functionally homogeneous but spatially distributed voxels, or, alternatively, as an anatomical landmark as those used in brain atlases (e.g. inferior frontal gyrus). However, observing functional regional selectivity in a distributed pattern is not incompatible with the observation of an absence of anatomical regional selectivity: a selective set of voxels may exist within an anatomically non-selective region. A second element concerns the choice of the stimuli. Some of the studies claiming functional selectivity used rather short auditory stimuli (*Boebinger et al., 2021*; *Norman-Haignere et al., 2015*; *Norman-Haignere et al., 2022*). Besides the low ecological validity of such stimuli that may reduce the generalizability of the findings (*Theunissen et al., 2000*), their comparison further relies on the assumption that speech and music share similar cognitive time constants. However, speech unfolds faster than music (*Ding et al., 2017*), and while a linguistic phrase is typically shorter than a second (*Inbar et al., 2020*), a melodic phrase is an order of magnitude longer. Moreover, balancing the complexity/simplicity of linguistic and musical stimuli can be challenging, and musical stimuli are often reduced to very simple melodies played on a synthesizer. These simple melodies mainly induce pitch processing in associative auditory regions (*Griffiths et al., 2010*) but do not recruit the entire dual-stream auditory pathways (*Zatorre et al., 2007*). Overall, while short and simple stimuli may be sufficient to induce linguistic processing, they might not be cognitively relevant musical stimuli. Finally, another element concerns the data at stake. Most studies that compared language and music processing examined functional MRI data (*Chen et al., 2023*; *Fedorenko et al., 2011*; *Nieto-Castañón and Fedorenko, 2012*). Here, we would like to consider cognition as resulting from interactions among functionally specialized but widely distributed brain networks and adopt an approach in which large-scale and frequency-specific neural dynamics are characterized. This approach rests on the idea that the canonical computations that underlie cognition and behavior are anchored in population dynamics of interacting functional modules (*Buzsáki and Vöröslakos, 2023*; *Safaie et al., 2023*) and bound to spectral fingerprints consisting of network- and frequency-specific coherent oscillations (*Siegel et al., 2012*). This framework requires relying on time-resolved neurophysiological recordings (M/EEG) and—rather than focusing only on the amplitude of the high-frequency activity (HFa), a common approach in the literature involving human intracranial EEG recordings (*Martin et al., 2019*; *Norman-Haignere et al., 2022*; *Oganian and Chang, 2019*)—to investigate the entire frequency spectrum of neural activity. Indeed, while HFa amplitude is a good proxy of focal neural spiking (*Le Van Quyen et al., 2010*; *Ray and Maunsell, 2011*), large-scale neuronal interactions mainly rely on slower dynamics (*Kayser et al., 2012*; *Kopell et al., 2000*; *Siegel et al., 2012*).

Following the reasoning developed above, we suggest that the study of selectivity of music and language processing should carefully consider the following points: First, the use of ecologically valid stimuli, both in terms of content and duration. Second, a within-subject approach comparing both conditions. Third, aiming for high spatial sensitivity. Fourth, considering not only one type of neural activity (broadband, HFa amplitude) but the entire frequency spectrum of the neurophysiological signal. Fifth, use a broad range of complementary analyses, including connectivity, and take into account individual variability. Finally, we suggest that terms should be operationally defined based on statistical tests, which results in a clear distinction between shared, selective, and preferred activity. That is, let A and B be two investigated cognitive functions, 'shared' would be a neural population that (compared to a baseline) significantly and equally contributes to the processing of both A and B;

'selective' would be a neural population that exclusively contributes to the processing of A or B (e.g. significant for A but not B); and 'preferred' would be a neural population that significantly contributes to the processing of both A and B, but more prominently for A or B (*Figure 1A*).

In an effort to take into account all the above challenges and to precisely quantify the degree of shared, preferred, and selective responses both at the levels of the channels and anatomical regions (*Figure 1C and D*), we conducted an experiment on 18 pharmacoresistant epileptic patients explored with stereotactic EEG (sEEG) electrodes. Patients listened to long and ecological audio-recordings of speech and music (10 min each). We investigated stimulus encoding, spectral content of the neural activity, and brain connectivity over the entire frequency spectrum (from 1 to 120 Hz; i.e. delta band to HFa). Finally, we carefully distinguished between the three different categories of neural responses described above: shared, selective, and preferred across the two investigated cognitive domains. Our results reveal that the majority of neural responses are shared between natural speech and music, and they highlight an absence of anatomical regional selectivity. Instead, we found neural selectivity to be restricted to distributed and frequency-specific coherent oscillations, typical of spectral fingerprints.

## Results

### Anatomical regional neural activity is mostly non-domain selective to speech or music

To investigate the presence of domain selectivity during ecological perception of speech and music, we first analyzed the neural responses to these two cognitive domains in both a spatially and spectrally resolved manner, with respect to two baseline conditions: one in which patients passively listened to pure tones (each 30 ms in duration), the other in which they passively listened to isolated syllables (/ba/ or /pa/, see Methods). Here, we will report the results using pure tones data as baseline, but note that the results using syllables data as baseline are highly similar (see *Figures 2–6* and corresponding figure supplements).We classified, for each canonical frequency band, each channel into one of the categories mentioned above, i.e., shared, selective, or preferred (*Figure 1A*), by examining whether speech and/or music differ from baseline and whether they differ from each other. We also considered both activations and deactivations, compared to baseline, as both index a modulation of neural population activity, and both have been linked with cognitive processes (*Pfurtscheller and Lopes da Silva, 1999*; *Proix et al., 2022*). However, because our aim was not to interpret specific increase or decrease with respect to the baseline, we here simply consider significant deviations from the baseline. In other words, when estimating selectivity, it is the strength of the response that matters, not its direction (activation, deactivation). Overall, neural responses are predominantly shared between the two domains, accounting for ~70% of the channels which showed a significant response compared to baseline (*Figures 2 and 3*). The preferred category is also systematically present, accounting for 3–15% of significant neural responses, across frequency bands. Selective responses are more present in the lower frequency bands (~30% up to the alpha band), and quite marginal in the HFa band (6–12%).

The spatial distribution of the spectrally resolved responses corresponds to the network typically involved in speech and music perception. This network encompasses both ventral and dorsal auditory pathways, extending well beyond the auditory cortex and hence beyond auditory processing that may result from differences in the acoustic properties of our baseline and experimental stimuli. This is the case for overall responses but also when only looking at shared responses. For instance, HFa shared responses represent 74–86% of the overall significant HFa responses, and are visible in the left superior and middle temporal gyri, inferior parietal lobule, and the precentral, middle, and inferior frontal gyri (*Figures 2F and 3F*). The left hemisphere appears to be more strongly involved, but this result is biased by the inclusion of a majority of patients with a left hemisphere exploration (*Figure 1C and D* and *Supplementary file 1*). Also, when inspecting left and right hemispheres separately, the patterns of shared, selective, and preferred responses remain similar across hemispheres across frequency bands (see *Figure 2—figure supplement 3* and *Figure 3—figure supplement 2* for activation and deactivation, respectively). Both domains displayed a comparable percentage of selective responses across frequency bands (*Figure 4*, first values of each plot). When considering separately activation (*Figure 2*) and deactivation (*Figure 3*) responses, speech and music showed complementary patterns: for low frequencies (<15 Hz) speech selective (and preferred) responses were mostly deactivations

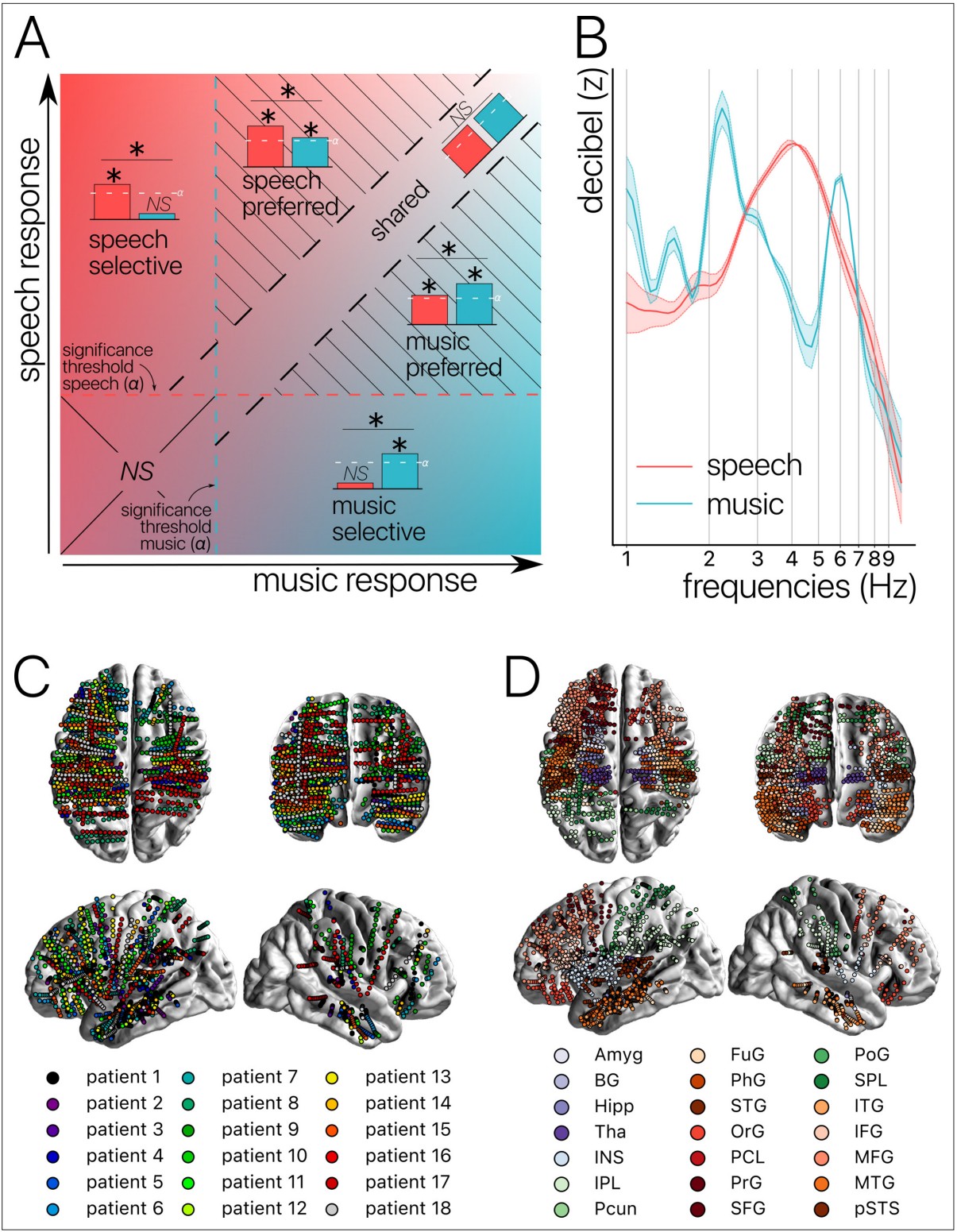

**Figure 1.** Concepts, stimuli, and recordings. (**A**) Conceptual definition of selective, preferred, and shared neural processes. Illustration of the continua between speech and music selectivity, speech and music preference, and shared resources. 'Selective' responses are neural responses significant for one domain but not the other, and with a significant difference between domains (for speech top left; for music bottom right). 'Preferred' responses correspond to neural responses that occur during both speech and music processing, but with a significantly stronger response for one domain over the other (striped triangles). Finally, 'shared' responses occur when there are no significant differences between domains, and there is a significant neural response to at least one of the two stimuli (visible along the diagonal). If neither domain produces a significant neural response, the difference is

*Figure 1 continued on next page*

*Figure 1 continued*

not assessed (lower left square). (**B**) Stimuli. Modulation spectrum of the acoustic temporal envelope of the continuous, 10 min long speech and music stimuli. (**C**) Anatomical localization of the stereotactic EEG (sEEG) electrodes for each patient (N=18). (**D**) Anatomical localization of the sEEG electrodes for each anatomical region. Abbreviations according to the Human Brainnetome Atlas (*Fan et al., 2016*).

and music responses activations compared to baseline, and this pattern reversed for high frequencies (>15 Hz).

Next, we investigated whether the channels selectivity (to speech or music) observed in a given frequency band was robust across frequency bands (*Figure 4*). We estimated the cross-frequency channel selectivity, that is the percentage of channels that selectively respond to speech or music across different frequency bands. We first computed the percentage of total channels selective for speech and music (either activated or deactivated compared to baseline) in a given frequency band. We then verified whether these channels were unresponsive to the other domain in the other frequency bands. This was done by examining each frequency band in turn and deducting any channels that showed a significant neural response to the other domain. When considering the entire frequency spectrum, the percentage of total channels being selective to speech or music is ~4 times less than when considering a single frequency band. For instance, while up to 8% of the total channels are selective for speech (or music) in the theta band, this percentage always drops to ~2% when considering the cross-frequency channel selectivity.

Critically, we found no evidence of anatomical regional selectivity, i.e., of a simple anatomo-functional spatial code (see *Figure 1D* for the definition of anatomical regions). We estimated, for each frequency band, activation/deactivation responses, and anatomical region, the proportion of patients showing selectivity for speech or music, by means of a population prevalence analysis (*Figures 5 and 6*; see Methods). This analysis revealed that, for the majority of patients, first of all, in most regions there were channels that responded to both speech and music (indicative of shared responses at the anatomical regional level), and, second of all, for the minority of anatomical regions for which a selectivity for the same domain (speech or music) was observed across multiple patients, this selectivity does not hold when also considering other frequency bands and activation/deactivation responses. For instance, while the left anterior middle temporal gyrus shows delta activity selective to music (*Figures 2A and 5A*), it shows low-gamma activity selective to speech (*Figures 2E and 5E*). The left superior temporal gyrus and pSTS (posterior superior temporal sulcus), which show selective activations in the theta and alpha bands for music (*Figure 5B and C*), show selective deactivations in the same bands for speech (*Figure 6B and C*) and a majority of shared activations in the HFa (*Figure 5F*). This absence of anatomical regional selectivity is also evident when looking at the uncategorized, continuous results (*Figure 2—figure supplement 2*).

Overall, these results reveal an absence of regional selectivity to speech or music under ecological conditions. Instead, selective responses coexist in space across different frequency bands. But, while selectivity may not be striking at the level of anatomical regional activity, it may still be present at the network level. To investigate this hypothesis, we explored the connectivity between the auditory cortex and the rest of the brain. And, to functionally define the auditory cortex for each patient, we first investigated the relation between the auditory signal itself and the brain response to identify which sEEG channels (spatial) best encode the dynamics of the auditory stimuli.

## Low-frequency neural activity best encodes acoustic dynamics

We linearly modeled the neurophysiological responses to continuous speech and music using temporal response functions (TRFs). Based on previous studies (*Oganian and Chang, 2019*; *Zion Golumbic et al., 2013*; *Zuk et al., 2021*), we compared four TRF models. From both stimuli, we extracted the continuous, broadband temporal envelope (henceforth 'envelope') and the discrete acoustic onset edges (henceforth 'peakRate'; see Methods) and we quantified how well these two acoustic features are encoded by either the low-frequency (LF) band (1–9 Hz) or the high-frequency amplitude (80–120 Hz) bands. For each model, we estimated the percentage of total channels for which a significant encoding was observed during speech and/or music listening. The model for which most channels significantly encoded speech and/or music acoustic features corresponded to the model in which LF neural activity encoded the *peakRates* (*Figure 7A*). In general, the LF activity encodes the acoustic features in significantly more channels than the HFa amplitude (*peakRate* & LF vs. *peakRate* & HFa

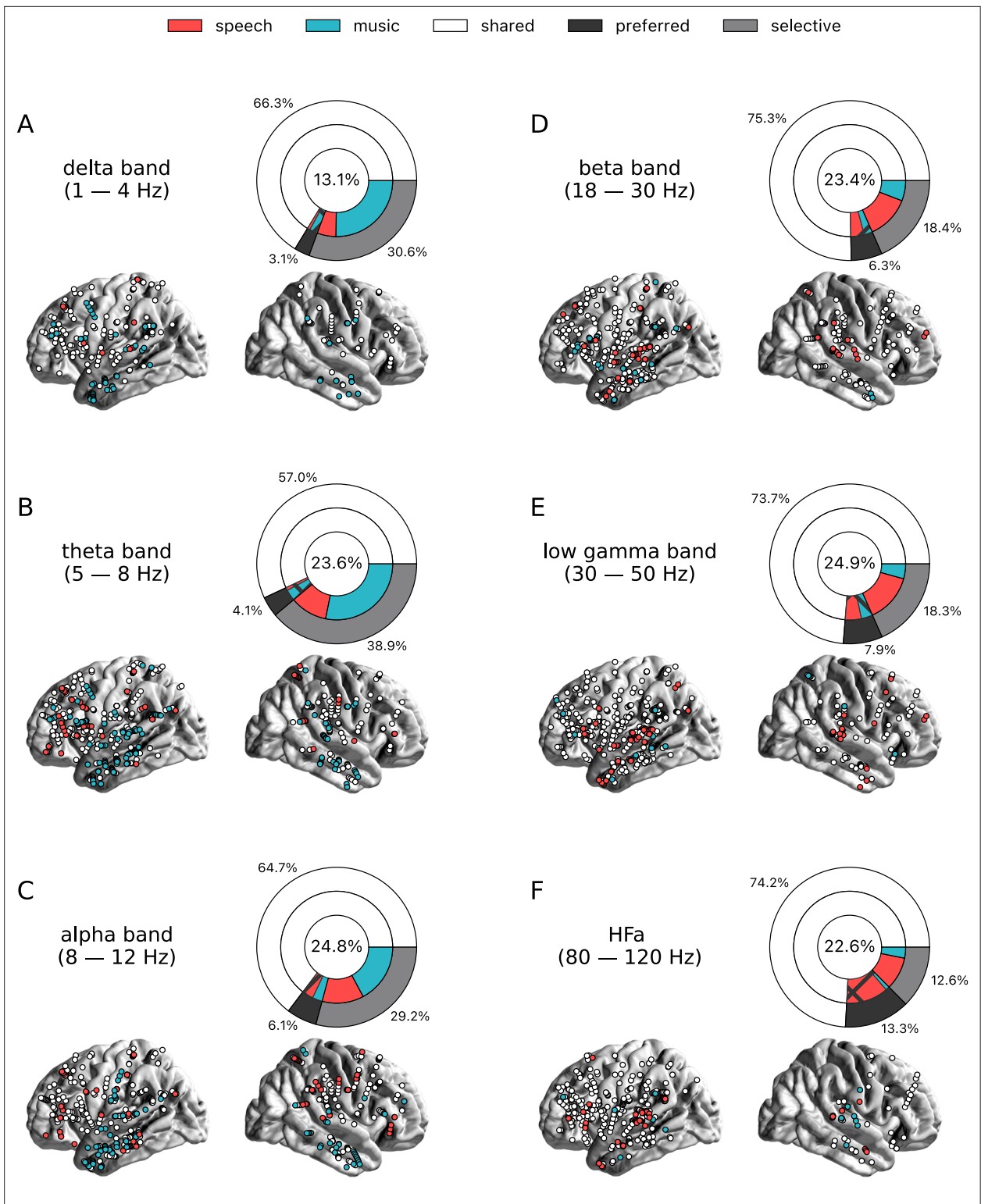

**Figure 2.** Power spectrum analyses of activations (speech or music>tones). (**A–F**) Neural responses to speech and/or music for the six canonical frequency bands. Only significant activations compared to the baseline condition (pure tones listening) are reported (see *Figure 2—figure supplement 2* for uncategorized, continuous results). Nested pie charts indicate: (1) in the center, the percentage of channels that showed a significant response to speech and/or music. (2) The outer pie indicates the percentage of channels, relative to the center, classified as shared (white), selective (light gray), and preferred (dark gray). (3) The inner pie indicates, for the selective (plain) and preferred (pattern) categories, the proportion of channels that were (more)

*Figure 2 continued on next page*

*Figure 2 continued*

responsive to speech (red) or music (blue). Brain plots indicate: Distribution of shared (white) and selective (red/blue) stereotactic EEG (sEEG) channels projected on the brain surface. Results are significant at q<0.01 (N=18).

The online version of this article includes the following figure supplement(s) for figure 2:

**Figure supplement 1.** Power spectrum analyses of activations (speech or music>syllables).

**Figure supplement 2.** Power spectrum analyses of activations (speech or music>tones) for each hemisphere separately and for the six frequency bands (**A-F**).

**Figure supplement 3.** Contrast between the neural responses to speech and music, for the six (**A-F**) canonical frequency bands (tones baseline).

---

amplitude comparison: t=13.39, q<0.0001; *peakRate* & LF vs. *envelope* & HFa amplitude comparison: t=9.55, q<0.0001). Note that this effect is not caused by the asymmetric comparison of bandpassed LF to HFa amplitude as model comparisons using the same extraction technique for both signals did not change the results (*Figure 7—figure supplement 1*). Then, while the *peakRates* are encoded by numerically more channels than the instantaneous envelope, this difference was not significant (*peakRate* & LF vs. *envelope* & LF comparison: t=1.93, q=0.42).

Furthermore, we show that the *peakRates* are encoded by the LF neural activity throughout the cortex, for both speech and music (*Figure 7B and C*). More precisely, the regions wherein neural activity significantly encodes the acoustic structure of the stimuli go well beyond auditory regions and extend to the temporo-parietal junction, motor cortex, inferior frontal gyrus, and anterior and central sections of the superior and middle temporal gyrus. In particular, the strongest encoding values for speech are observed in the typical left-hemispheric language network, comprising the upper bank of the superior temporal gyrus, the posterior part of the inferior frontal gyrus, and the premotor cortex (*Malik-Moraleda et al., 2022*). Still, as expected, the best cortical tracking of the acoustic structure takes place in the auditory cortex, for both speech and music (*Figure 7D*). In other words, the best encoding channels are the same for speech and music and are those located closest to—or in—the primary auditory cortex. While the left hemisphere appears to be more strongly involved, this result is biased by the inclusion of a majority of patients with a left hemisphere exploration (see *Figure 1C and D* and *Supplementary file 1*). Proportionally, we found no difference in the number of significant channels between hemispheres (i.e. speech: 41% and 44% for left and right hemispheres respectively; music: 22% and 24% for left and right hemispheres, respectively). Finally, the *peakRate* & LF model, i.e., the model that captures the largest proportion of significant channels during speech and/or music perception (*Figure 7A*), yields for both classes of stimuli a similar TRF shape (*Figure 7E*) as well as similar prediction accuracy scores (Pearson's r), of up to 0.55 (*Figure 7F*).

## Connections of the auditory cortex are also mostly non-domain selective to speech or music

Seed-based connectivity analyses first revealed that, during speech or music perception, the auditory cortex is mostly connected to the rest of the brain through slow neural dynamics, with ~33% of the channels showing coherence values higher than the surrogate distribution at delta rate, and only ~12% at HFa (*Figure 8*, see also *Figure 8—figure supplement 1* for uncategorized, continuous results). Across frequencies, most of the significant connections are shared between the two cognitive domains (~70%), followed by preferred (~15%) and selective connections (~12%). Selectivity is nonetheless homogeneously present in all frequency bands (*Figure 8*). Importantly, selectivity is again frequency-specific (*Figure 9*). Estimating the cross-frequency channel selectivity, the percentage of total connections being selective to speech or music is at zero for all frequency bands except for the delta range (speech = 0.19%; music = 0.06%). Hence, selectivity is only visible at the level of frequency-specific distributed networks. Finally, here again no anatomical regional selectivity is observed, i.e., not a single cortical region is solely selective to speech or music. Rather, in every cortical region, the majority of patients show shared responses at the regional level, as estimated by the population prevalence analysis (*Figure 10*).

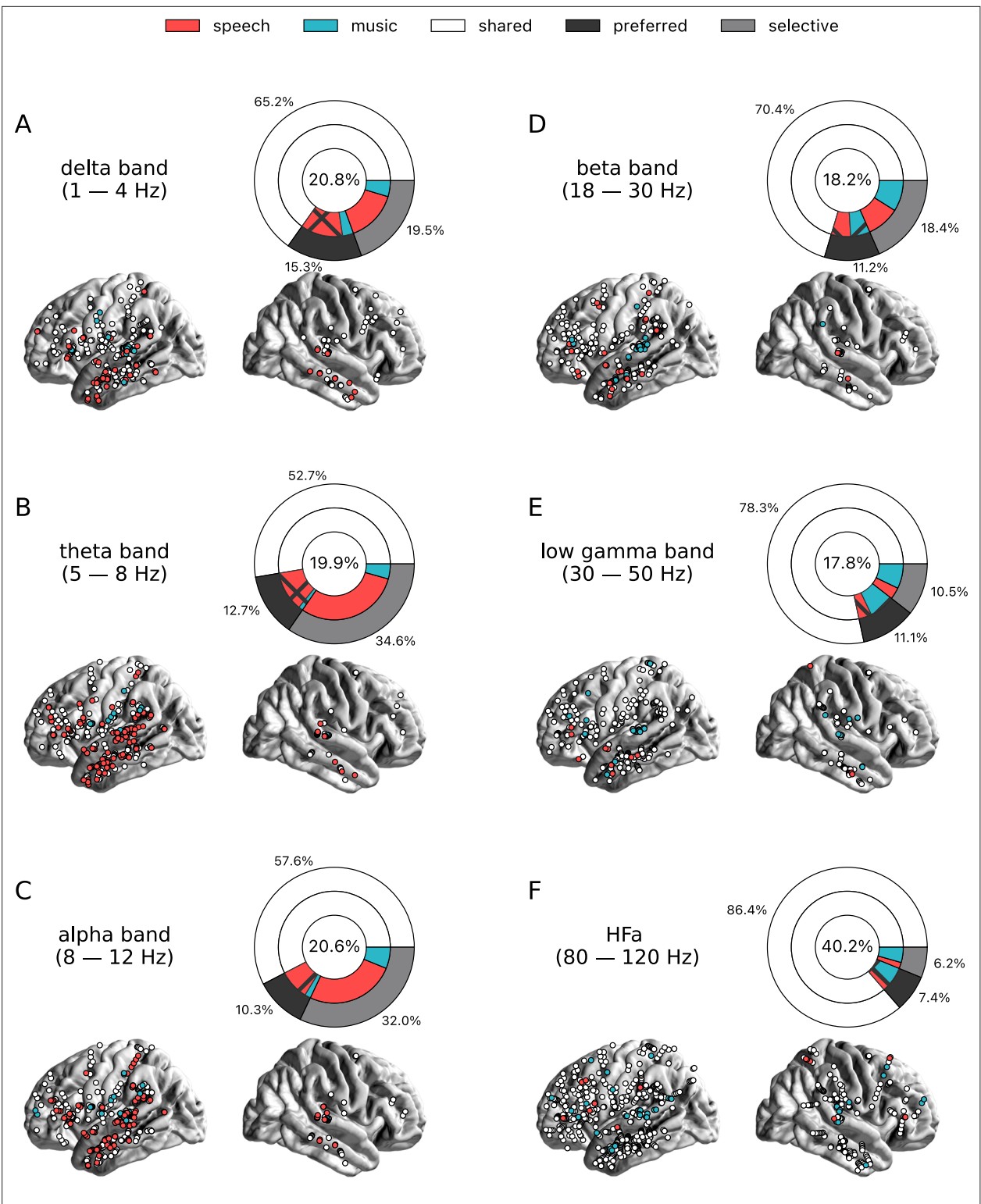

**Figure 3.** Power spectrum analyses of deactivations (speech or music<tones). (**A–F**) Neural responses to speech and/or music for the six canonical frequency bands. Only significant deactivations compared to the baseline condition (pure tones listening) are reported. Same conventions as in *Figure 2*. Results are significant at q<0.01 (N=18).

The online version of this article includes the following figure supplement(s) for figure 3:

**Figure supplement 1.** Power spectrum analyses of deactivations (speech or music<syllables).

**Figure supplement 2.** Power spectrum analyses of deactivations (speech or music<tones) for each hemisphere separately.

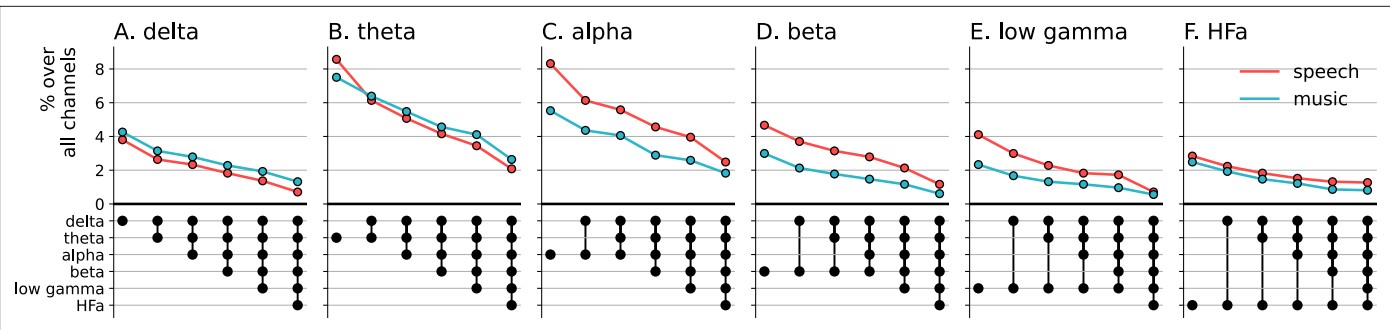

**Figure 4.** Cross-frequency channel selectivity for the power spectrum analyses. (**A-F**) Percentage of channels that exclusively respond to speech (red) or music (blue) across different frequency bands. For each plot, the first (leftmost) value corresponds to the percentage (%) of channels displaying a selective response in a specific frequency band (either activation or deactivation, compared to the baseline condition of pure tones listening). In the next value, we remove the channels that are significantly responsive in the other domain (i.e. no longer exclusive) for the following frequency band (e.g. in panel **A**: speech selective in delta; speech selective in delta XOR music responsive in theta; speech selective in delta XOR music responsive in theta XOR music responsive in alpha; and so forth). The black dots at the bottom of the graph indicate which frequency bands were successively included in the analysis. Note that channels remaining selective across frequency bands did not necessarily respond selectively in every band. They simply never showed a significant response to the other domain in the other bands.

The online version of this article includes the following figure supplement(s) for figure 4:

**Figure supplement 1.** Cross-frequency channel selectivity for the power spectrum analyses (syllables baseline).

## Discussion

In this study, we investigated the existence of domain selectivity for speech and music under ecological conditions. We capitalized on the high spatiotemporal sensitivity of human stereotactic recordings (sEEG) to thoroughly evaluate the presence of selective neural responses—estimated both at the level of individual sEEG channels and anatomical cortical regions—when patients listened to a story or to instrumental music. More precisely, we statistically quantified the extent to which natural speech and music processing is performed by shared, preferred, or domain-selective neural populations. By combining sEEG investigations of HFa with the analyses of other frequency bands (from delta to low-gamma), the neural encoding of acoustic dynamics and spectrally resolved connectivity analyses, we obtained a thorough characterization of the neural dynamics at play during natural and continuous speech and music perception. Our results show that speech and music mostly rely on shared neural resources. Further, while selective responses seem absent at the level of atlas-based cortical regions, selectivity can be observed at the level of frequency-specific distributed networks in both power and connectivity analyses.

Previous work has reported that written or spoken language selectively activates a left-lateralized functional cortical network (*Chen et al., 2023*; *Fedorenko et al., 2011*; *Fedorenko and Blank, 2020*; *Malik-Moraleda et al., 2022*). In particular, in previous functional MRI studies, these strong and selective cortical responses were not visible during the presentation of short musical excerpts, and are hypothesized to index linguistic processes (*Chen et al., 2023*; *Fedorenko et al., 2011*). Moreover, in the superior temporal gyrus, specific and separate neural populations for speech, music, and song are visible (*Boebinger et al., 2021*; *Norman-Haignere et al., 2022*). These selective responses, not visible in primary cortical regions, seem independent of both low-level acoustic features and higher-order linguistic meaning (*Norman-Haignere et al., 2015*), and could subtend intermediate representations (*Giordano et al., 2023*) such as domain-dependent predictions (*McCarty et al., 2023*; *Sankaran et al., 2023*). Within this framework, the localizationism view applies to highly specialized processes (i.e. functional niches), while general cognitive domains are mostly spatially distributed. Recent studies have shown that some communicative signals (e.g. alarm, emotional, linguistic) can exploit distinct acoustic niches to target specific neural networks and trigger reactions adapted to the intent of the emitter (*Albouy et al., 2020*; *Arnal et al., 2019*). Using neurally relevant spectro-temporal representations (MPS), these studies show that different subspaces encode distinct information types: slow temporal modulations for meaning (speech), fast temporal modulations for alarms (screams), and spectral modulations for melodies (*Albouy et al., 2020*; *Arnal et al., 2015*; *Arnal et al., 2019*; *Flinker et al., 2019*). Which acoustic features—and which neural mechanisms—are necessary and sufficient

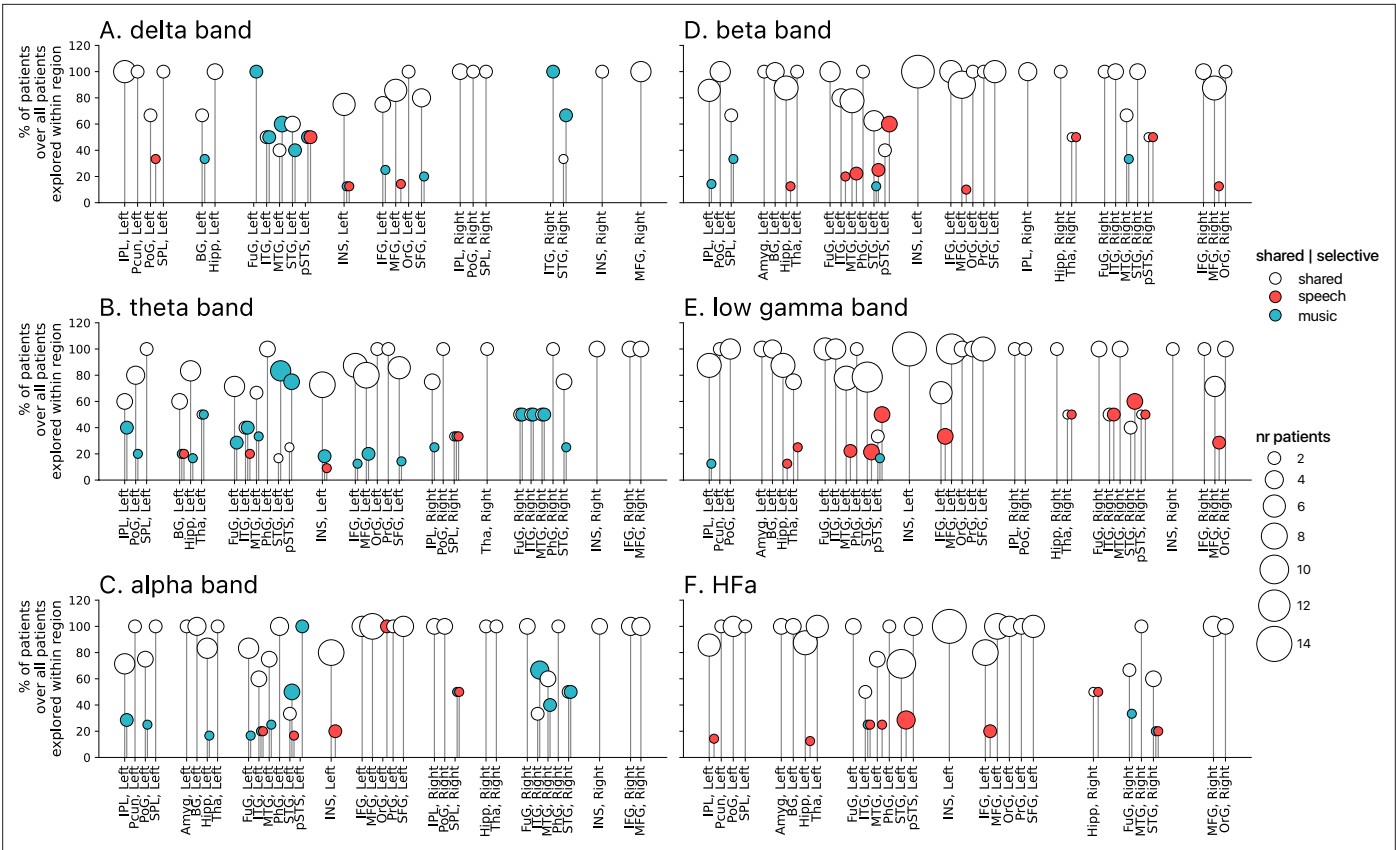

**Figure 5.** Population prevalence for the power spectral analyses of activations (speech or music>tones; N=18). (**A–F**) Population prevalence of shared or selective responses for the six canonical frequency bands, per anatomical region (note that preferred responses are excluded). Only significant activations compared to the baseline condition (pure tones listening) are reported. Regions (on the x-axis) were included in the analyses if they had been explored by minimally two patients with minimally two significant channels. Patients were considered to show regional selective processing when all their channels in a given region responded selectively to either speech (red) or music (blue). When regions contained a combination of channels with speech selective, music selective, or shared responses, the patient was considered to show shared (white) processing in this region. The height of the lollipop (y-axis) indicates the percentage of patients over the total number of explored patients in that given region. The size of the lollipop indicates the number of patients. As an example, in panel F (high-frequency activity [HFa] band), most lollipops are white with a height of 100%, indicating that, in these regions, all patients presented a shared response profile. However, in the left inferior parietal lobule (IPL, left) one patient (out of the seven explored) shows speech selective processing (filled red circle). A fully selective region would thus show a fixed-color full height across all frequency bands. Abbreviations according to the Human Brainnetome Atlas (**Fan et al., 2016**).

The online version of this article includes the following figure supplement(s) for figure 5:

**Figure supplement 1.** Population prevalence for the power spectral analyses of activations (speech or music>syllables; N=18).

to route communicative sounds toward selective neural networks remains a promising field of investigation to explore.

In this context, in the current study we did not observe a single anatomical region for which speech selectivity was present, in any of our analyses. In other words, 10 min of instrumental music was enough to activate cortical regions classically labeled as speech (or language)-selective. On the contrary, we report spatially distributed and frequency-specific patterns of shared, preferred, or selective neural responses and connectivity fingerprints. This indicates that domain-selective brain regions should be considered as a set of functionally homogeneous but spatially distributed voxels, instead of anatomical landmarks. Several non-exclusive explanations may account for this finding. First, our results part with the simple selective versus shared dichotomy and adopt a more biologically valid and continuous framework (**Buzsáki, 2019**; **Zatorre and Gandour, 2008**) by adding a new category that is often neglected in the literature: *preferred* responses (**Figure 1A**). Indeed, responses in this category are usually reported as shared or selective and most often the statistical approach does not allow a more nuanced view (**Chen et al., 2023**). However, preferred responses, namely responses that are stronger

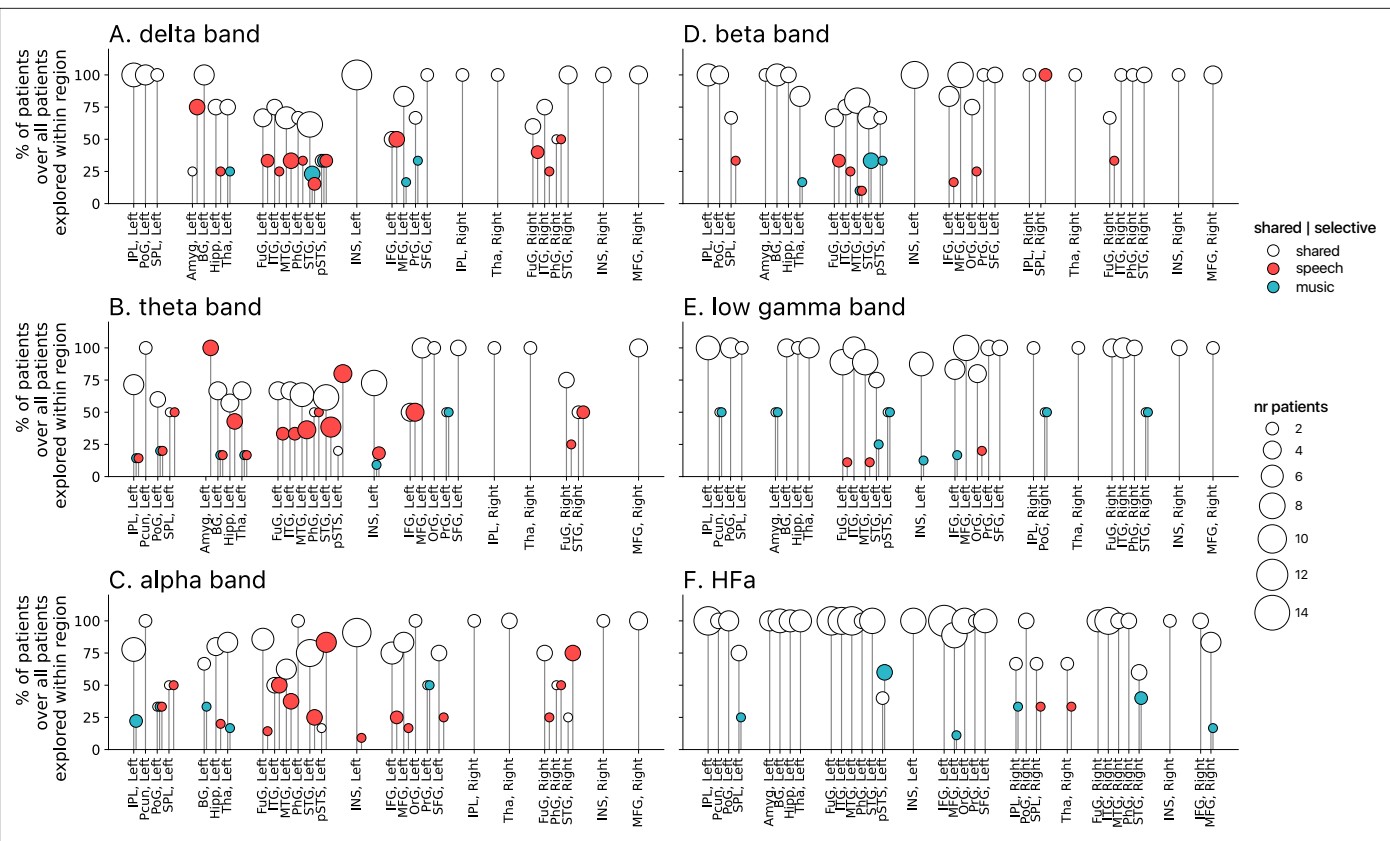

**Figure 6.** Population prevalence for the power spectral analyses of deactivations (speech or music<tones; N=18). (**A–F**) Population prevalence of shared or selective responses for the six canonical frequency bands, per anatomical region. Only significant deactivations compared to the baseline condition (pure tones listening) are reported. Same conventions as in *Figure 5*.

The online version of this article includes the following figure supplement(s) for figure 6:

**Figure supplement 1.** Population prevalence for the power spectral analyses of deactivations (speech or music<syllables; N=18).

to a given class of stimuli but that are also present with other stimuli, are relevant and should not be collapsed into either the selective or shared categories. Introducing this intermediate category refines the epistemological and statistical approach on how to map cognitive and brain functions. It points toward the presence of gradients of neural activity across cognitive domains, instead of all-or-none responses. This framework is more compatible with the notion of distributed representations wherein specific regions are more or less recruited depending on their relative implication in a distributed manifold (*Elman, 1991*; *Rissman and Wagner, 2012*).

Second, most of the studies that reported regional selectivity are grounded on functional MRI data that lack a precise temporal resolution. Furthermore, the few studies assessing selectivity with intracranial EEG recordings analyzed only the HFa amplitude (*Bellier et al., 2022*; *Norman-Haignere et al., 2020*; *Oganian and Chang, 2019*). However, while this latter reflects local (*Kopell et al., 2000*) and possibly feedforward activity (*Bastos et al., 2015*; *Fontolan et al., 2014*; *Fries, 2015*), other frequency bands are also constitutive of the cortical dynamics and involved in cognition. For instance, alpha/beta rhythms play a role in predicting upcoming stimuli and modulating sensory processing and associated spiking (*Arnal and Giraud, 2012*; *Bastos et al., 2020*; *Morillon and Baillet, 2017*; *Saleh et al., 2010*; *van Kerkoerle et al., 2014*). Also slower dynamics in the delta/theta range have been described to play a major role in cognitive processes and in particular for speech perception, contributing to speech tracking, segmentation, and decoding (*Ding et al., 2017*; *Doelling et al., 2014*; *Giraud and Poeppel, 2012*; *Gross et al., 2013*; *Keitel et al., 2018*). Importantly, we here addressed both activations and deactivations that can co-occur in the same spatial location across different frequency bands (*Pfurtscheller and Lopes da Silva, 1999*; *Proix et al., 2022*) and indeed observed that the domain selectivity observed within our restricted stimulus set is frequency-specific, meaning

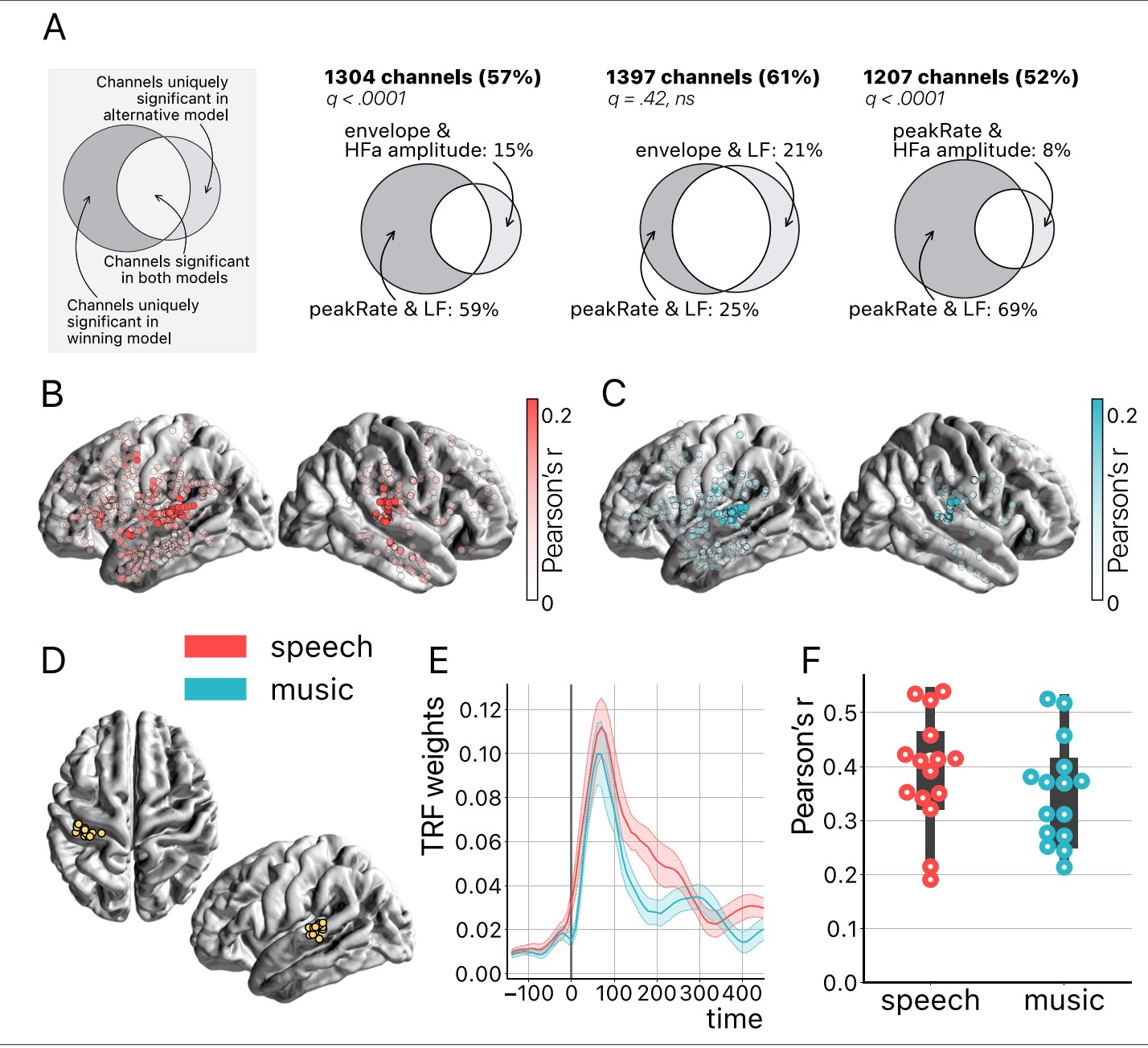

**Figure 7.** Temporal response function (TRF) analyses. (**A**) Model comparison. On the top left a toy model illustrates the use of Venn diagrams comparing the winning model (peakRate in low frequency [LF]) to each of the three other models for speech and music (pooled). Four TRF models were investigated to quantify the encoding of the instantaneous envelope and the discrete acoustic onset edges (peakRate) by either the LF band or the high-frequency amplitude. The 'peakRate & LF' model significantly captures the largest proportion of channels, and is, therefore, considered the winning model. The percentages on the top (in bold) indicate the percentage of total channels for which a significant encoding was observed during speech and/or music listening in either of the two compared models. In the Venn diagram, we indicate, out of all significant channels, the percentage that responded in the winning model (left) or in the alternative model (right). The middle part indicates the percentage of channels shared between the winning and the alternative model (percentage not shown). q-Values indicate pairwise model comparisons (Wilcoxon signed-rank test, FDR-corrected). (**B and C**) peakRate & LF model: Spatial distribution of stereotactic EEG (sEEG) channels wherein LF neural activity significantly encodes the speech (red) and music (blue) peakRates. Higher prediction accuracy (Pearson's r) is indicated by increasing color saturation. All results are significant at q<0.01 (N=18). (**D**) Anatomical localization of the best encoding channel within the left hemisphere for each patient (N=15), as estimated by the 'peakRate & LF' model (averaged across speech and music). These channels are all within the auditory cortex and serve as seeds for subsequent connectivity analyses. (**E**) TRFs averaged across the seed channels (N=15), for speech and music. (**F**) Prediction accuracy (Pearson's r) of the neural activity of each seed channel, for speech and music.

*Figure 7 continued on next page*

*Figure 7 continued*

The online version of this article includes the following figure supplement(s) for figure 7:

**Figure supplement 1.** Temporal response function (TRF) model comparison of low-frequency (LF) amplitude and high-frequency activity (HFa) amplitude.

---

that domain selectivity is marginal when considering the entire spectrum of activity of a given sEEG channel. Finally, most studies only investigated local neural activity and did not consider the brain as a distributed system, analyzed through the lens of functional connectivity analyses. While topological approaches are more complex, they also provide more nuanced and robust characterization of brain functions. Critically, our approach reveals the limitation of adopting a reductionist approach—either by considering the brain as a set of independent regions instead of distributed networks, or by over-looking the spectral complexity of the neural signal.

Third, the ecological auditory stimuli we used are longer and more complex than stimuli used in previous studies and hence more prone to elicit distributed and dynamical neural responses (*Hasson et al., 2010*; *Sonkusare et al., 2019*; *Theunissen et al., 2000*) and they require, in the case of music, for instance, more complex representations of melody and rhythm motifs contributing to stronger representations of meter, tonality, and groove (*Boebinger et al., 2021*). While listening to natural speech and music rests on cognitively relevant neural processes, our analytical approach, extending over a rather long period of time, does not allow to directly isolate specific brain operations. Computational models—which can be as diverse as acoustic (*Chi et al., 2005*), cognitive (*Giordano et al., 2021*), information-theoretic (*Di Liberto et al., 2020*; *Donhauser and Baillet, 2020*), or self-supervised neural networks (*Donhauser and Baillet, 2020*; *Millet et al., 2022*; *Sankaran et al., 2023*) models—are hence necessary to further our understanding of the type of computations performed by our reported frequency-specific distributed networks. Moreover, incorporating models accounting for musical and linguistic structure can help us avoid misattributing differences between speech and music driven by unmatched sensitivity factors (e.g. arousal, emotion, or attention) as inherent speech or music selectivity (*Mas-Herrero et al., 2013*; *Nantais and Schellenberg, 1999*).

Our modeling approach, although lacking the modeling of melodic and linguistic features, was targeting the temporal dynamics of the speech and music stimuli. Beyond confirming that acoustic dynamics are strongly tracked by auditory neural dynamics, it revealed, investigating the entire cortex, that such neural tracking also occurs well outside of auditory regions—up to motor and inferior frontal areas (*Figure 7B*; see also *Chalas et al., 2022*; *Zion Golumbic et al., 2013*). Of note, this spatial map of speech dynamics encoding is very similar to former reports of the brain regions belonging to the language system (*Diachek et al., 2020*). But, here again, adopting an approach that investigates both low and high frequencies of the neural signal—an approach that is not enough embraced in intra-cranial EEG studies (*Proix et al., 2022*)—reveals that the LF activity clearly better encodes acoustic features than the HFa amplitude (*Figure 7A*).

In conclusion, our results point to a massive amount of shared neural response to speech and music, well beyond the auditory cortex. They also show the interest of considering shared, preferred, and selective responses when investigating domain selectivity. Importantly these three classes of responses should be considered in respect to (1) activation or deactivation patterns compared to a baseline, (2) different frequency bands, and (3) power spectrum (activity) and connectivity approaches. Combining all these points of view gives a richer although possibly more complex view of brain functions. While our data point to an absence of anatomical regional selectivity for speech and music, such a selectivity still exists at the level of a spatially distributed and frequency-specific network. Thus, the inconsistency with previous findings may be limited to the idea that some anatomical regions are selective to speech or music processing. However, the two points of view can be reconciled when considering a fine-grained network approach allowing selectivity to coexist for speech and music within the same anatomical region. Finally, in adopting here a comparative approach of speech and music—the two main auditory domains of human cognition—we only investigated one type of speech and of music during a passive listening task. Future work is needed to investigate for instance whether different sentences or melodies activate the same selective frequency-specific distributed networks and to what extent these results are related to the passive listening context compared to a more active and natural context (e.g. conversation).

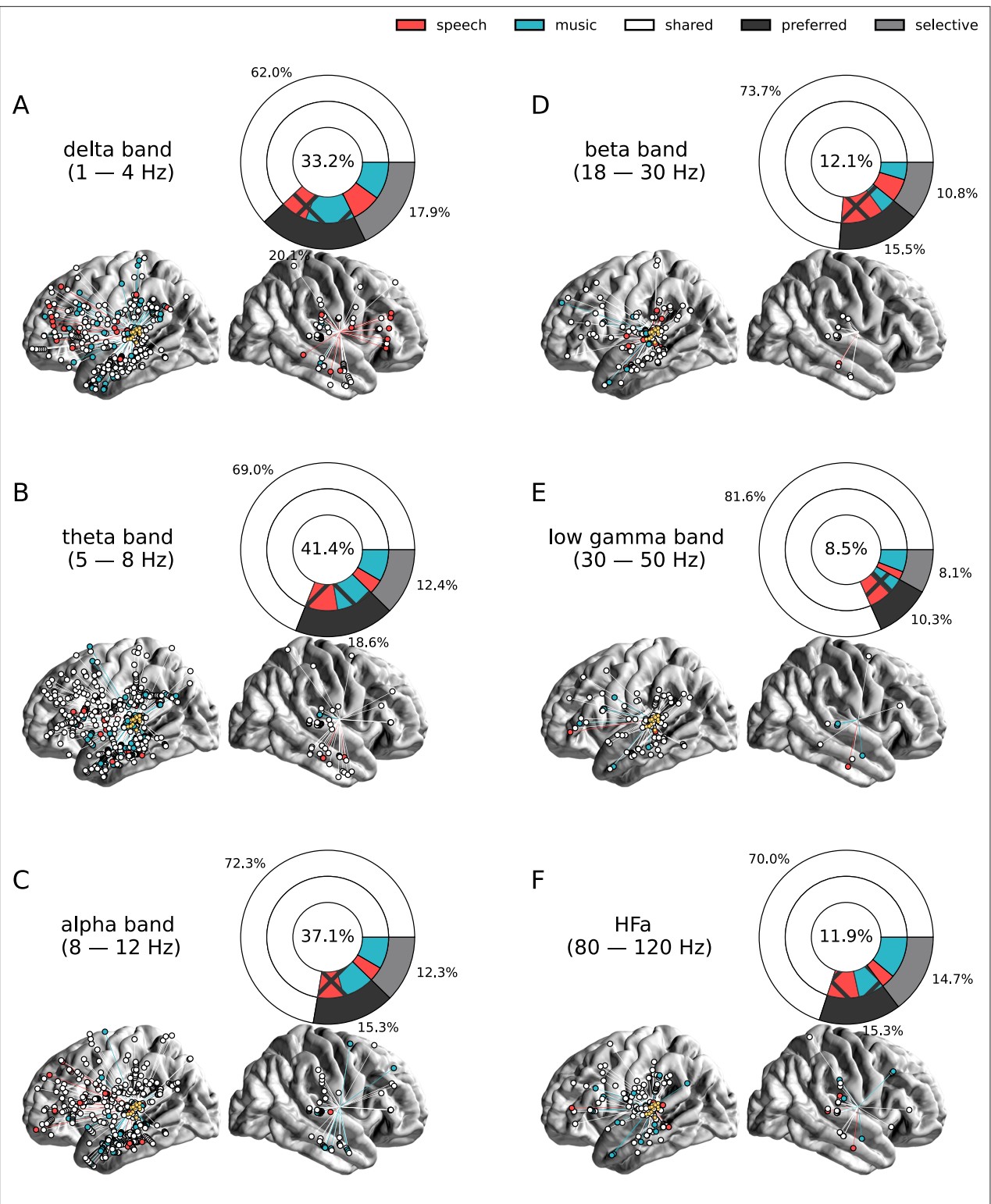

**Figure 8.** Seed-based functional connectivity analyses. (**A–F**) Significant coherence responses to speech and/or music for the six canonical frequency bands (see *Figure 8—figure supplement 1* for uncategorized, continuous results). The seed was located in the left auditory cortex (see *Figure 7D*). Same conventions as in *Figure 2*, except for the center percentage in the nested pie charts which, here, reflects the percentage of channels significantly connected to the seed. Results are significant at q<0.01 (N=15).

The online version of this article includes the following figure supplement(s) for figure 8:

**Figure supplement 1.** Contrast between the coherence responses to speech and music, for the six canonical frequency bands (**A-F**).

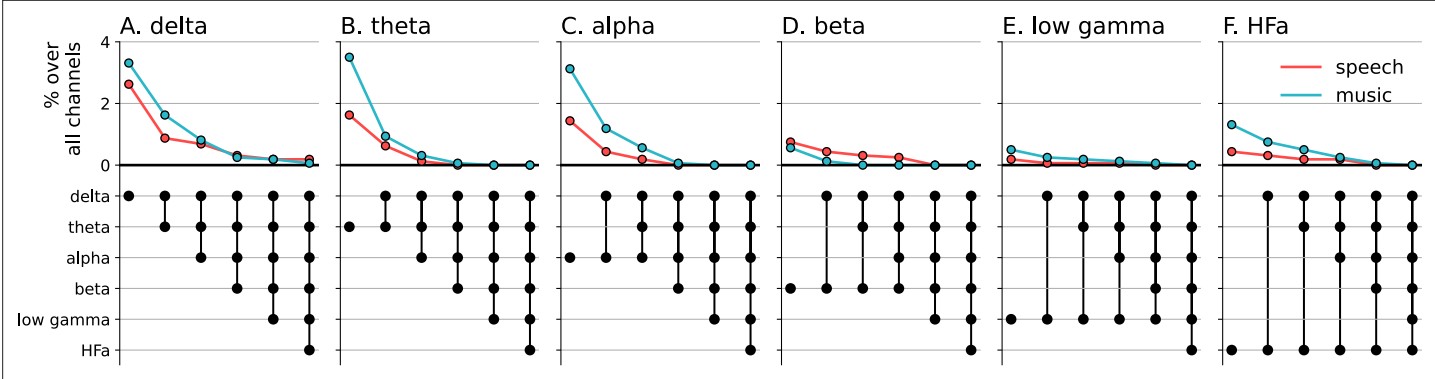

**Figure 9.** Cross-frequency channel selectivity for the connectivity analyses. Percentage of channels that showed selective coherence with the primary auditory cortex in speech (red) or music (blue) across different frequency bands (**A-F**). Same conventions as in *Figure 4*.

# Methods

## Participants

18 patients (10 females, mean age 30 years, range 8–54 years) with pharmacoresistant epilepsy participated in the study. All patients were French native speakers. Neuropsychological assessments carried out before sEEG recordings indicated that all patients had intact language functions and met the criteria for normal hearing. In none of them were the auditory areas part of their epileptogenic zone as identified by experienced epileptologists. Recordings took place at the Hôpital de La Timone (Marseille, France). Patients provided informed consent prior to the experimental session, and the experimental protocol was approved by the Institutional Review Board of the French Institute of Health (IRB00003888).

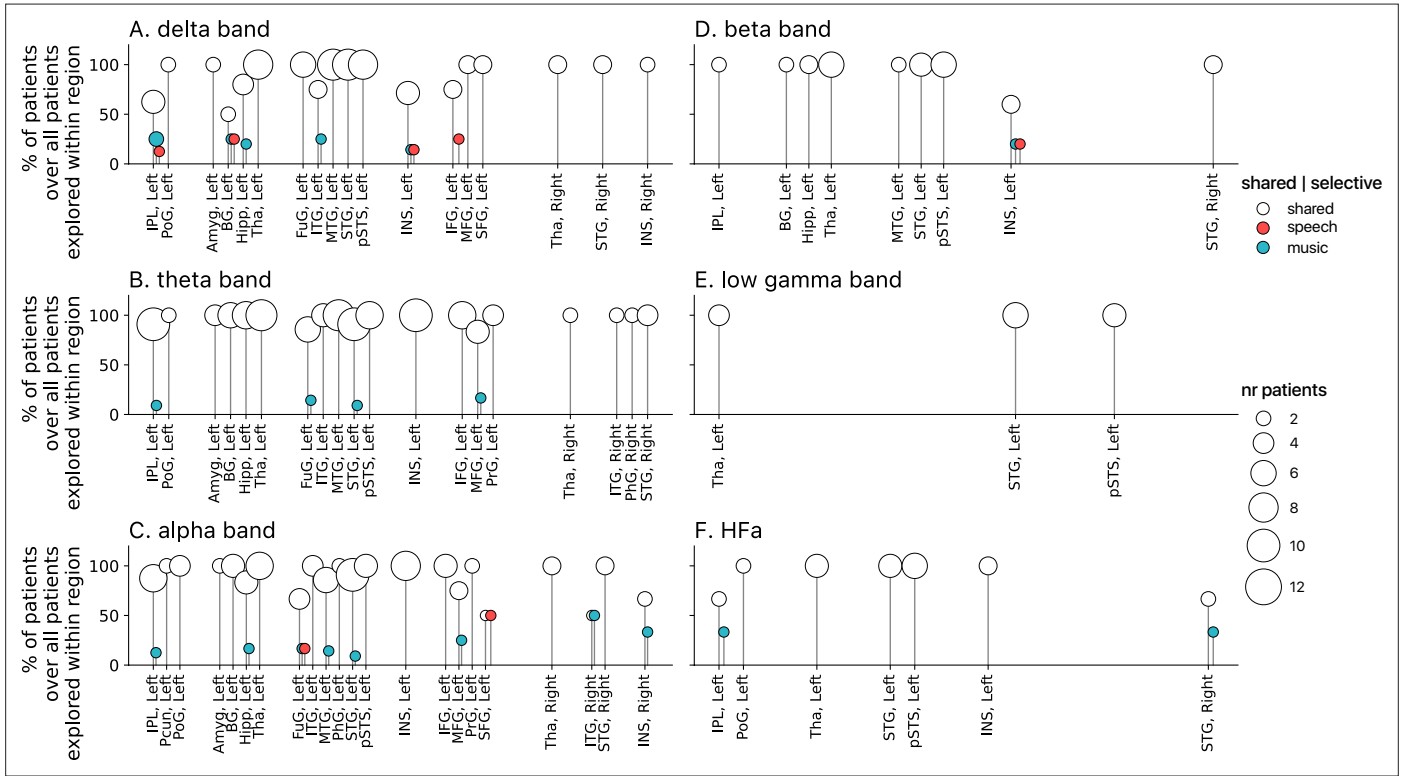

**Figure 10.** Population prevalence for the connectivity analyses for the six (**A-F**) canonical frequency bands (N=15). Same conventions as in *Figure 5*.

## Data acquisition

The sEEG signal was recorded using depth electrodes shafts of 0.8 mm diameter containing 10–15 electrode contacts (Dixi Medical or Alcis, Besançon, France). The contacts were 2 mm long and were spaced from each other by 1.5 mm. The locations of the electrode implantations were determined solely on clinical grounds. Patients were included in the study if their implantation map covered at least partially the Heschl's gyrus (left or right). The cohort consists of 13 unilateral implantations (10 left, 3 right) and 5 bilateral implantations, yielding a total of 271 electrodes and 3371 contacts (see *Figure 1C and D* for electrodes localization).

Patients were recorded either in an insulated Faraday cage or in the bedroom. In the Faraday cage, they laid comfortably in a chair, the room was sound attenuated, and data were recorded using a 256-channel amplifier (Brain Products), sampled at 1 kHz and high-pass filtered at 0.016 Hz. In the bedroom, data were recorded using a 256-channel Natus amplifier (DeltaMed system), sampled at 512 Hz, and high-pass filtered at 0.16 Hz.

## Experimental design

Patients completed three separate sessions. In one session they completed the main experimental paradigm and the two additional sessions served as baseline for the spectral analysis (see below).

In the main experimental session, patients passively listened to ~10 min of storytelling (*Gripari, 2004*); 577 s, *La sorcière de la rue Mouffetard* (*Gripari, 2004*) and ~10 min of instrumental music (580 s, *Reflejos del Sur, Oneness, 2006*) separated by 3 min of rest. The order of conditions was counterbalanced across patients (see *Supplementary file 1*). This session was conducted in the Faraday cage (N=6) or in the bedroom (N=12).

In the two baseline sessions, patients passively listened to two more basic types of auditory stimuli: (1) 30-ms-long pure tones, presented binaurally at 500 Hz or 1 kHz (with a linear rise and fall time of 0.3 ms) 110 times each, with an ISI of 1030 (±200) ms; and (2) /ba/ or /pa/ syllables, pronounced by a French female speaker and presented binaurally 250 times each, with an ISI of 1030 (±200) ms. These stimuli were designed for a clinical purpose in order to functionally map the auditory cortex. These two recording sessions (lasting ~2 and 4 min) were performed in the Faraday cage.

In the Faraday cage, a sound Blaster X-Fi Xtreme Audio, an amplifier Yamaha P2040 and Yamaha loudspeakers (NS 10M) were used for sound presentation. In the bedroom, stimuli were presented using a Sennheiser HD 25 headphone set. Sound stimuli were presented at 44.1 kHz sample rate and 16 bits resolution. Speech and music excerpts were presented at ~75 dBA (see *Figure 1B*).

## General preprocessing related to electrodes localization

To increase spatial sensitivity and reduce passive volume conduction from neighboring regions (*Mercier et al., 2017*; *Mercier et al., 2022*), the signal was offline re-referenced using bipolar montage. That is, for a pair of adjacent electrode contacts, the referencing led to a virtual channel located at the midpoint locations of the original contacts. To precisely localize the channels, a procedure similar to the one used in the iELVis toolbox and in the FieldTrip toolbox was applied (*Groppe et al., 2017*; *Stolk et al., 2018*). First, we manually identified the location of each channel centroid on the post-implant CT scan using the Gardel software (*Medina Villalon et al., 2018*). Second, we performed volumetric segmentation and cortical reconstruction on the pre-implant MRI with the Freesurfer image analysis suite (documented and freely available for download online at http://surfer.nmr.mgh.harvard.edu/). This segmentation of the pre-implant MRI with SPM12 provides us with both the tissue probability maps (i.e. gray, white, and cerebrospinal fluid [CSF] probabilities) and the indexed-binary representations (i.e. either gray, white, CSF, bone, or soft tissues). This information allowed us to reject electrodes not located in the brain. Third, the post-implant CT scan was coregistered to the pre-implant MRI via a rigid affine transformation and the pre-implant MRI was registered to MNI152 space, via a linear and a non-linear transformation from SPM12 methods (*Penny et al., 2011*), through the FieldTrip toolbox (*Oostenveld et al., 2011*). Fourth, applying the corresponding transformations, we mapped channel locations to the pre-implant MRI brain that was labeled using the volume-based Human Brainnetome Atlas (*Fan et al., 2016*).

Based on the brain segmentation performed using SPM12 methods through the FieldTrip toolbox, bipolar channels located outside of the brain were removed from the data (3%). The remaining data (*Figure 1C*) was then bandpass filtered between 0.1 and 250 Hz, and, following a visual inspection of

the power spectral density profile of the data, when necessary, we additionally applied a notch filter at 50 Hz and harmonics up to 200 Hz to remove power line artifacts (N=12). Finally, the data were downsampled to 500 Hz.

## Artifact rejection

To define artifacted channel we used both the broadband signal and the amplitude of the HFa. This latter was obtained by computing, with the Hilbert transform, the analytic amplitude of four 10-Hz-wide sub-bands spanning from 80 to 120 Hz. Each sub-band was standardized by dividing it by its mean and, finally, all sub-bands were averaged together (*Ossandón et al., 2012*; *Vidal et al., 2012*). Channels with a variance greater than 2*IQR (interquartile range, i.e. a non-parametric estimate of the standard deviation)—on either the broadband or high-frequency signals—were tagged as artifacted channels (on average 18% of the channels). Then the data were epoched in non-overlapping segments of 5 s (2500 samples). To exclude artifacted epochs, epochs, wherein the maximum amplitude (over time) summed across non-excluded channels was greater than 2*IQR, were tagged as artifacted epochs. Overall, 6% of the speech epochs and 7% of the music epochs were rejected. Channels and epochs defined as artifacted were excluded from subsequent analyses, except if specified otherwise (see TRF analysis section).

## Spectral analysis

Six canonical frequency bands were investigated: delta (1–4 Hz), theta (5–8 Hz), alpha (8–12 Hz), beta (18–30 Hz), low-gamma (30–50 Hz), and HFa (80–120 Hz). To prevent edge artifacts, prior to extracting the power spectrum, epochs were zero-padded on both sides with 3.5 s segments which were later removed. For each patient, channel, epoch, and frequency band, the power of the neural signal was calculated using the Welch approach on discrete Fourier transform from the SciPy-Python library (*Virtanen et al., 2020*) and then averaged across the relevant frequencies to obtain these six canonical bands.

For each canonical band and each channel, we classified the time-averaged neural response as being selective, preferred, or shared across the two investigated cognitive domains (speech, music). We defined these categories by capitalizing on both the simple effects of—and contrast between— the neural responses to speech and music stimuli compared to a baseline condition (see *Figure 1A*). 'Selective' responses are neural responses that are significantly different compared to the baseline for one domain (speech or music) but not the other, and with a significant difference between domains (i.e. speech or music is different from baseline+difference effect between the domains). 'Preferred' responses correspond to neural responses that occur during both speech and music processing, but with a significantly stronger response for one domain over the other (i.e. both speech and music are significantly different from baseline+difference effect between the domains). Finally, 'shared' responses occur when there are no significant differences between domains, and there is a significant neural response to at least one of the two stimuli (one or two simple effects+no difference). If none of the two domains produces a significant neural response, the difference is not assessed (case 'neither' simple effect). In order to explore the full range of possible selective, preferred, or shared responses, we considered both responses greater and smaller than the baseline. Indeed, as neural populations can synchronize or desynchronize in response to sensory stimulation, we estimated these categories separately for significant activations and significant deactivations compared to baseline.

For each frequency band and channel, the statistical difference between conditions was estimated with paired sample permutation tests based on the t-statistic from the MNE-Python library (*Gramfort et al., 2014*) with 1000 permutations and the *tmax* method to control the family-wise error rate (*Groppe et al., 2011*; *Nichols and Holmes, 2002*). In *tmax* permutation testing, the null distribution is estimated by, for each channel (i.e. each comparison), swapping the condition labels (speech vs music or speech/music vs baseline) between epochs. After each permutation, the most extreme t-scores over channels (*tmax*) are selected for the null distribution. Finally, the t-scores of the observed data are computed and compared to the simulated *tmax* distribution, similar as in parametric hypothesis testing. Because with an increased number of comparisons, the chance of obtaining a large *tmax* (i.e. false discovery) also increases, the test automatically becomes more conservative when making more comparisons, as such correcting for the multiple comparison between channels.

## TRF analysis

We used the TRF to estimate the encoding of acoustic features by neural activity. Two acoustic features were extracted from our stimuli (speech, music): the envelope and the peakRate. To estimate the temporal envelope of the two stimuli, the acoustic waveforms were decomposed into 32 narrow frequency bands using a cochlear model, and the absolute value of the Hilbert transform was computed for each of these narrowband signals. The broadband temporal envelope (henceforth 'envelope') resulted from the summation of these absolute values. The acoustic onset edges (henceforth 'peakRate') were defined as peaks in the rate of change (first derivative) of the envelope (*Doelling et al., 2014*; *Oganian and Chang, 2019*). Finally, both the envelopes and peakRates were downsampled to 100 Hz and z-scored to be mapped to the neural data.

All computations of the TRF used the pymTRF library (*Steinkamp, 2019*), a Python adaption of the mTRF toolbox (*Crosse et al., 2016*). A TRF is a model that, via linear convolution, serves as a filter to quantify the relationship between two continuous signals, here stimulus features and neural activity. Hence, for this analysis, the entire duration of the recordings were preserved, i.e., no artifacted epochs were excluded. When applied in a forward manner, the TRF approach describes the mapping of stimulus features onto the neural response (henceforth 'encoding'; *Crosse et al., 2016*). Using ridge regression to avoid overfitting, we examined how well the two different acoustic features—envelope and peakRate—map onto LF activity (1–9 Hz) or the amplitude of the HFa (80–120 Hz, see Artifact rejection section) (*Ding et al., 2016*; *Zion Golumbic et al., 2013*). Hence, four encoding models were estimated: envelope/peakRate acoustic features * LF/amplitude of HFa neural activity. For each model and patient, the optimal ridge regularization parameter ($\lambda$) was estimated using cross-validation on the sEEG channels situated in the auditory cortex. We considered time lags from –150 to 1000 ms for the TRF estimations. 80% of the data was used to derive the TRFs and the remaining 20% was used as a validation set. The quality of the predicted neural response was assessed by computing Pearson's product moment correlations (Fisher z-scored) between the predicted and actual neural data for each channel and model using the SciPy-Python library (p-values FDR-corrected).

Models were finally compared in terms of the percentage of channels that significantly encoded the acoustic structure of speech and/or music. This percentage was estimated at the single-subject level and combined with non-parametric Wilcoxon signed-rank tests at the group level to define the winning model. In other words, the winning model is the model for which the percentage of channels significantly encoding speech and/or music acoustic features is the largest. Multiple comparison across pairs of models was controlled for with a FDR correction.

## Connectivity analysis

We examined the frequency-specific functional connectivity maps in response to speech and music, between the entire brain and the auditory cortex using a seed-based approach (we dismissed the channels immediately neighboring the seed channel). As seed, we selected, per patient, the channel that best encoded the speech and music acoustic features (see TRF analysis; *Figure 7D*). We used spectral coherence as a connectivity measure for all canonical bands (see above) and all analyses were performed using the MNE-Python library (*Gramfort et al., 2014*). Our rationale to use coherence as functional connectivity metric was threefold. First, coherence analysis considers both magnitude and phase information. While the absence of dissociation can be criticized, signals with higher amplitude and/or SNR lead to better time-frequency estimates (which is not the case with a metric that would focus on phase only and therefore would be more likely to include estimates of various SNR). Second, we choose a metric that allows direct comparison between frequencies. As, at high frequencies phase angle changes more quickly, phase alignment/synchronization is less likely in comparison with lower frequencies. Third, we intend to align to previous work which, for the most part, used the measure of coherence most likely for the reasons explained above.

For each frequency band, we classified each channel into selective, preferred, or shared categories (see *Figure 1A*) by examining both the simple effects (i.e. which channels display a significantly coherent signal with the seed during speech and/or music processing) and the difference effects (i.e. is coherence significantly stronger for one domain over the other).

Statistical significance was assessed for each frequency band and channel using surrogate data with 1000 iterations, which were generated by modifying the temporal structure of the sEEG signal recorded at the seeds (i.e. shuffling the epochs) prior to computing connectivity. This process led to

a total of 1000 connectivity values, which were used as null distribution to calculate the probability threshold associated with genuine connectivity.

## Population prevalence

For both the spectral and the connectivity analyses, in order to make sure that the results are not driven by the heterogeneity of electrode locations across patients, we examined, for each region, the proportions of patients showing only shared or selective responses. That is, for both the spectral and connectivity results, we examined results representativeness as follows: for each anatomical region wherein at least two patients have at least two significantly responsive channels, we computed the percentage of patients that showed a pattern of selective (i.e. all channels selective to speech or music) or a shared (i.e. a mixture of channels responding to speech and/or music) responses. This approach is inspired by the population prevalence, where an equivalent metric is introduced (i.e. the maximum a posterior estimate; see *Ince et al., 2021*).

## Acknowledgements

We thank all patients for their willingful participation. We thank Patrick Marquis for helping with the data acquisition, and Anne-Catherine Tomei and all colleagues from the Institut de Neuroscience des Systèmes for useful discussions. ANR-20-CE28-0007-01 (to BM), ANR-21-CE28-0010 (to DS), ANR-17-EURE-0029 (NeuroMarseille), and co-funded by the European Union (ERC, SPEEDY, ERC-CoG-101043344). This work, carried out within the Institute of Convergence ILCB, was also supported by grants from France 2030 (ANR-16-CONV-0002), the French government under the Programme 'Investissements d'Avenir', and the Excellence Initiative of Aix-Marseille University (A*MIDEX, AMX-19-IET-004).

## Additional information

### Funding

| Funder | Grant reference number | Author |
|---|---|---|
| Agence Nationale de la Recherche | ANR-21-CE28-0010 | Daniele Schön |
| Agence Nationale de la Recherche | ANR-20-CE28-0007-01 | Benjamin Morillon |
| Agence Nationale de la Recherche | ANR-17-EURE-0029 | Benjamin Morillon Daniele Schön |
| European Research Council | ERC-CoG-101043344 | Benjamin Morillon |
| Agence Nationale de la Recherche | ANR-16-CONV-0002 | Benjamin Morillon Daniele Schön |
| Aix-Marseille Université | A*MIDEX AMX-19-IET-004 | Benjamin Morillon Daniele Schön |

The funders had no role in study design, data collection and interpretation, or the decision to submit the work for publication.

### Author contributions

Noémie te Rietmolen, Formal analysis, Visualization, Methodology, Writing – original draft, Writing – review and editing; Manuel R Mercier, Data curation, Formal analysis, Methodology, Writing – original draft, Writing – review and editing; Agnès Trébuchon, Data curation, Writing – review and editing; Benjamin Morillon, Conceptualization, Formal analysis, Visualization, Methodology, Writing – original draft, Project administration, Writing – review and editing; Daniele Schön, Conceptualization, Formal analysis, Funding acquisition, Visualization, Methodology, Writing – original draft, Project administration, Writing – review and editing

## Author ORCIDs
Noémie te Rietmolen (ID) https://orcid.org/0000-0002-5532-6118
Manuel R Mercier (ID) http://orcid.org/0000-0001-6358-4734
Benjamin Morillon (ID) https://orcid.org/0000-0002-0049-064X
Daniele Schön (ID) http://orcid.org/0000-0003-4472-4150

## Ethics

Human subjects: Patients provided informed consent prior to the experimental session, and the experimental protocol was approved by the Institutional Review board of the French Institute of Health (IRB00003888).

Reviewer #1 (Public review): https://doi.org/10.7554/eLife.94509.3.sa1
Reviewer #2 (Public review): https://doi.org/10.7554/eLife.94509.3.sa2
Reviewer #3 (Public review): https://doi.org/10.7554/eLife.94509.3.sa3
Author response https://doi.org/10.7554/eLife.94509.3.sa4

---

# Additional files

## Supplementary files

• Supplementary file 1. Patients description. Table provides an overview of patient characteristics and experimental conditions. The table includes 18 patients, with a mix of males and females aged between 8 and 54 years (mean 30). Presentation order (either speech-music or music-speech) was counterbalanced between patients. Recordings took place either at the bedside (room) or in the lab. Hemispheric dominance was mostly typical. All patients had an electrode implanted in the auditory cortex (Heschl's gyrus; left, right, or bilaterally). The number of depth electrodes is indicated in the far right column.

• MDAR checklist

## Data availability

The raw data investigated in the current manuscript is privileged patient data. Because of this, the conditions of our ethics approval do not permit public archiving of anonymised study data. Readers seeking access to the data should contact Dr. Daniele Schön (daniele.schon@univ-amu.fr). Access will be granted to named individuals in accordance with ethical procedures governing the reuse of clinical data, including completion of a formal data sharing agreement. Data analyses were performed using custom scripts in Python, which are available on GitHub along with preprocessed data necessary to reproduce the figures and results: https://github.com/noemietr/iSpeech (copy archived *te Rietmolen, 2024*).

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
