## [Editor Report · eLife assessment]

This study presents **valuable** intracranial findings on how two types of natural auditory stimuli - speech and music - are processed in the human brain, and demonstrates that speech and music largely share network-level brain activities, thus challenging the domain-specific processing view. The evidence supporting the claims of the authors is **solid**. The work will be of broad interest to speech and music researchers as well as cognitive scientists in general.

---

## [Referee Report · Reviewer #1 (Public review)]

Summary:

In this study, the authors examined the extent to which processing of speech and music depends on neural networks that are either specific to a domain or general in nature. They conducted comprehensive intracranial EEG recordings on 18 epilepsy patients as they listened to natural, continuous forms of speech and music. This enabled an exploration of brain activity at both the frequency-specific and network levels across a broad spectrum. Utilizing statistical methods, the researchers classified neural responses to auditory stimuli into categories of shared, preferred, and domain-selective types. It was observed that a significant portion of both focal and network-level brain activity is commonly shared between the processing of speech and music. However, neural responses that are selectively responsive to speech or music are confined to distributed, frequency-specific areas. The authors highlight the crucial role of using natural auditory stimuli in research and the need to explore the extensive spectral characteristics inherent in the processing of speech and music.

Strengths:

The study's strengths include its high-quality sEEG data from a substantial number of patients, covering a majority of brain regions. This extensive cortical coverage grants the authors the ability to address their research questions with high spatial resolution, marking an advantage over previous studies. They performed thorough analyses across the entire cortical coverage and a wide frequency range of neural signals. The primary analyses, including spectral analysis, temporal response function calculation, and connectivity analysis, are presented straightforwardly. These analyses, as well as figures, innovatively display how neural responses, in each frequency band and region/electrode, are 'selective' (according to the authors' definition) to speech or music stimuli. The findings are summarized in a manner that efficiently communicates information to readers. This research offers valuable insights into the cortical selectivity of speech and music processing, making it a noteworthy reference for those interested in this field. Overall, this research offers a valuable dataset and carries out extensive yet clear analyses, amounting to an impressive empirical investigation into the cortical selectivity of speech and music. It is recommended for readers who are keen on understanding the nuances of selectivity and generality in the processing of speech and music to refer to this study's data and its summarized findings.

Weaknesses:

(1) The study employed longer speech and music stimuli, thereby promising improved ecological validity as compared to prior research, a point emphasized by the authors. However, it failed to differentiate between neural responses to the diverse content or local structures within speech and music. The authors considered the potential limitation of treating these extensive speech and music stimuli as stationary signals, neglecting their complex musical or linguistic structural details and temporal variations across local structures such as sentences and phrases. This balanced perspective offered by the authors aids readers in better understanding the context of the study and highlights potential areas for expansion and further considerations.

(2) In contrast to previous studies that employed short stimulus segments along with various control stimuli to ensure that observed selectivity for speech or music was not merely due to low-level acoustic properties, this study used longer, ecological stimuli. However, the control stimuli used in this study, such as tone or syllable sequences, do not align with the low-level acoustic properties of the speech and music stimuli. This mismatch raises concerns that the differences or selectivity between speech and music observed in this study might be attributable to these basic acoustic characteristics rather than to more complex processing factors specific to speech or music. However, this should not deter readers from recognizing the study's strengths, namely, the use of iEEG recordings that offer high spatial resolution and extensive cortical coverage.

(3) The concept of selectivity - shared, preferred, and domain-selective - may not present sufficient theoretical accuracy. It is appreciated that the authors put effort into clearly defining their operational measurement on 'selectivity'. Later, the authors further mentioned the specific indication of their analyses. However, the authors' categorization of neural sites/regions as shared, preferred, or domain-selective regarding speech and music processing essentially resembles a traditional ANOVA test with posthoc analysis. While this categorization gives meaningful context to the results, the mere presence of significant differences among control stimuli, a segment of speech, and a piece of music does not present a strong case that a region is specifically selective to a type of stimulus like speech. The narrative of the manuscript could potentially lead to an overgeneralized interpretation of their findings as being broadly applicable to speech or music, if a reader does not delve into the details.

(4) The authors' approach, akin to mapping a 'receptive field' by correlating stimulus properties with neural responses to ascertain functional selectivity for speech and music, presents potential issues. If cortical regions exhibit heightened responses to one type of stimulus over another, it doesn't automatically imply selectivity or preference for that stimulus. The explanation could lie in functional aspects, such as a region's sensitivity to temporal units of a specific duration, be it music, speech, or even movie segments, and its role in chunking such units (e.g., around 500 ms), which might be more prevalent in music than in speech, or vice versa in the current study. This study does not delve into the functional mechanisms of how speech and music are processed across different musical or linguistic hierarchical levels but merely demonstrates differences in neural responses to various stimuli over a 10-minute span.

---

## [Referee Report · Reviewer #2 (Public review)]

Summary:

The study investigates whether speech and music processing involve specific or shared brain networks. Using intracranial EEG recordings from 18 epilepsy patients, it examines neural responses to speech and music. The authors found that most neural activity is shared between speech and music processing, without specific regional brain selectivity. Furthermore, domain-selective responses to speech or music are limited to frequency-specific coherent oscillations. The findings challenge the notion of anatomically distinct regions for different cognitive functions in the auditory process.

Strengths:

(1) This study uses a relatively large corpus of intracranial EEG data, which provides high spatiotemporal resolution neural recordings, allowing for more precise and dynamic analysis of brain responses. The use of continuous speech and music enhances ecological validity compared to artificial or segmented stimuli.

(2) This study uses multiple frequency bands in addition to just high-frequency activity (HFA), which has been the focus of many existing studies in the literature. This allows for a more comprehensive analysis of neural processing across the entire spectrum. The heterogeneity across different frequency bands also indicates that different frequency components of the neural activity may reflect different underlying neural computations.

(3) This study also adds empirical evidence towards distributed representation versus domain-specificity. It challenges the traditional view of highly specialized, anatomically distinct regions for different cognitive functions. Instead, the study suggests a more integrated and overlapping neural network for processing complex stimuli like speech and music.

Weaknesses:

While this study is overall convincing, there are still some weaknesses in the methods and analyses that limit the implication of the work.

The study's main approach, focusing primarily on the grand comparison of response amplitudes between speech and music, may overlook intricate details in neural coding. Speech and music are not entirely orthogonal with each other at different levels of analysis: at the high-level abstraction, these are two different categories of cognitive processes; at the low-level acoustics, they overlap a lot; at intermediate levels, they may also share similar features. For example, the study doesn't adequately address whether purely melodic elements in music correlate with intonations in speech at the neural level. A more granular analysis, dissecting stimuli into distinct features like pitch, phonetics, timbre, and linguistic elements, could unveil more nuanced shared, and unique neural processes between speech and music. Prior research indicates potential overlap in neural coding for certain intermediate features in speech and music (Sankaran et al. 2023), suggesting that a simple averaged response comparison might not fully capture the complexity of neural encoding. Further delineation of phonetic, melodic, linguistic, and other coding, along with an analysis of how different informational aspects (phonetic, linguistic, melodic, etc) are represented in shared neural activities, could enhance our understanding of these processes and strengthen the study's conclusions.

While classifying electrodes into 3 categories provides valuable insights, it may not fully capture the complexity of the neural response distribution to speech and music. A more nuanced and continuous approach could reveal subtler gradations in neural response, rather than imposing categorical boundaries. This could be done by computing continuous metrics, like unique variances explained by each category or by each acoustic feature, etc. Incorporating such a continuum could enhance our understanding of the neural representation of speech and music, providing a more detailed and comprehensive picture of cortical processing. This goes back to my first comment that the selected set of stimuli may not fully exploit the entire space of speech and music, and there are possible exemplars that violate the preference map here. For example, this study only considered a specific set of multi-instrumental music, it is not clear to me if other types of music would result in different response profiles in individual channels. It is also not clear if a foreign language that the listeners cannot comprehend would evoke similar response profiles. On the contrary, breaking down into the neural coding of more fundamental feature representations that constitute speech and music, and analyzing the unique contribution of each feature would give a more comprehensive understanding.

The paper's emphasis on shared and overlapping neural activity, as observed through sEEG electrodes, provides valuable insights. It is probably true that domain-specificity for speech and music does not exist at such a macro scale. However, it's important to consider that each electrode records from a large neuronal population, encompassing thousands of neurons. This broad recording scope might mask more granular, non-overlapping feature representations at the single neuron level. Thus, while the study suggests shared neural underpinnings for speech and music perception at a macroscopic level, it cannot definitively rule out the possibility of distinct, non-overlapping neural representations at the microscale of local neuronal circuits for features that are distinctly associated with speech and music. This distinction is crucial for fully understanding the neural mechanisms underlying speech and music perception that merit future endeavors with more advanced large-scale neuronal recordings.

---

## [Referee Report · Reviewer #3 (Public review)]

Summary:

Te Rietmolen et al., investigated the selectivity of cortical responses to speech and music stimuli using neurosurgical stereo EEG in humans. The authors address two basic questions: 1. Are speech and music responses localized in the brain or distributed; 2. Are these responses selective and domain specific or rather domain general and shared. To investigate this, the study proposes a nomenclature of shared responses (speech and music responses are not significantly different), domain selective (one domain is significant from baseline and the other is not), domain preferred (both are significant from baseline but one is larger than the other and significantly different from each other). The authors employ this framework using neural responses across the spectrum (rather than focusing on high gamma), providing evidence for a low level of selectivity across spectral signatures. To investigate the nature of the underlying representations they use encoding models to predict neural responses (low and high frequency) given a feature space of the stimulus envelope or peak rate (by time delay) and find stronger encoding for both in the low frequency neural responses. The top encoding electrodes are used as seeds for a pair-wise connectivity (coherence) in order to repeat the shared/selective/preferred analysis across the spectra, suggesting low selectivity. Spectral power and connectivity are also analyzed on the level of regional patient population to rule out (and depict) any effects driven by a select few patients. Across analyses the authors consistently show a paucity of domain selective responses and when evident these selective responses were not represented across the entire cortical region. The authors argue that speech and music mostly rely on shared neural resources.

Strengths:

I found this manuscript to be rigorous providing compelling and clear evidence towards shared neural signatures for speech and music. The use of intracranial recordings provides an important spatial and temporal resolution that lends itself to the power, connectivity and encoding analyses. The statistics and methods employed are rigorous and reliable, estimated based on permutation approaches and cross-validation/regularization was employed and reported properly. The analysis of measures across the entire spectra in both power, coherence and encoding models provides a comprehensive view of responses that no doubt will benefit the community as an invaluable resource. Analysis on the level of patient population (feasible with their high N) per region also supports the generalizability of the conclusions across a relatively large cohort of patients. Last but not least, I believe the framework of selective, preferred, and shared is a welcome lens through which to investigate cortical function.

Weaknesses:

I did not find methodological weaknesses in the current version of the manuscript. I do believe that it is important to highlight that the data is limited to passively listening to naturalistic speech and music. The speech and music stimuli are not completely controlled with varying key acoustic features (inherent to the different domains). Overall, I found the differences in stimulus and lack of attentional controls (passive listening) to be minor weaknesses that would not dramatically change the results or conclusions.

---

## [Author Response]

The following is the authors’ response to the original reviews.

We have specifically addressed the points of uncertainty highlighted in eLife's editorial assessment, which concerned the lack of low-level acoustics control, limitations of experimental design, and in-depth analysis. Regarding “the lack of low-level acoustics control, limitations of experimental design”, in response to Reviewer #1, we clarify that our study aimed to provide a broad perspective —which includes both auditory and higher-level processes— on the similarities and distinctions in processing natural speech and music within an ecological context. Regarding “the lack of in-depth analysis”, in response to Reviewer #1 and #2, we have clarified that while model-based analyzes are valuable, they pose fundamental challenges when comparing speech and music. Non-acoustic features inherently differ between speech and music (such as phonemes and pitch), making direct comparisons reliant on somewhat arbitrary choices. Our approach mitigates this challenge by analyzing the entire neural signal, thereby avoiding potential pitfalls associated with encoding models of non-comparable features. Finally, we provide some additional analyzes suggested by the Reviewers.

We sincerely appreciate your thoughtful and thorough consideration throughout the review process.

**eLife assessment**
This study presents valuable intracranial findings on how two important types of natural auditory stimuli - speech and music - are processed in the human brain, and demonstrates that speech and music largely share network-level brain activities, thus challenging the domain-specific processing view. The evidence supporting the claims of the authors is solid but somewhat incomplete since although the data analysis is thorough, the results are robust and the stimuli have ecological validity, important considerations such as low-level acoustics control, limitations of experimental design, and in-depth analysis, are lacking. The work will be of broad interest to speech and music researchers as well as cognitive scientists in general.
**Reviewer #1 (Public Review):**
Summary:In this study, the authors examined the extent to which the processing of speech and music depends on neural networks that are either specific to a domain or general in nature. They conducted comprehensive intracranial EEG recordings on 18 epilepsy patients as they listened to natural, continuous forms of speech and music. This enabled an exploration of brain activity at both the frequency-specific and network levels across a broad spectrum. Utilizing statistical methods, the researchers classified neural responses to auditory stimuli into categories of shared, preferred, and domain-selective types. It was observed that a significant portion of both focal and network-level brain activity is commonly shared between the processing of speech and music. However, neural responses that are selectively responsive to speech or music are confined to distributed, frequency-specific areas. The authors highlight the crucial role of using natural auditory stimuli in research and the need to explore the extensive spectral characteristics inherent in the processing of speech and music.Strengths:The study's strengths include its high-quality sEEG data from a substantial number of patients, covering a majority of brain regions. This extensive cortical coverage grants the authors the ability to address their research questions with high spatial resolution, marking an advantage over previous studies. They performed thorough analyses across the entire cortical coverage and a wide frequency range of neural signals. The primary analyses, including spectral analysis, temporal response function calculation, and connectivity analysis, are presented straightforwardly. These analyses, as well as figures, innovatively display how neural responses, in each frequency band and region/electrode, are 'selective' (according to the authors' definition) to speech or music stimuli. The findings are summarized in a manner that efficiently communicates information to readers. This research offers valuable insights into the cortical selectivity of speech and music processing, making it a noteworthy reference for those interested in this field. Overall, this research offers a valuable dataset and carries out extensive yet clear analyses, amounting to an impressive empirical investigation into the cortical selectivity of speech and music. It is recommended for readers who are keen on understanding the nuances of selectivity and generality in the processing of speech and music to refer to this study's data and its summarized findings.Weaknesses:The weakness of this study, in my view, lies in its experimental design and reasoning:(1) Despite using longer stimuli, the study does not significantly enhance ecological validity compared to previous research. The analyses treat these long speech and music stimuli as stationary signals, overlooking their intricate musical or linguistic structural details and temporal variation across local structures like sentences and phrases. In previous studies, short, less ecological segments of music were used, maintaining consistency in content and structure. However, this study, despite employing longer stimuli, does not distinguish between neural responses to the varied contents or structures within speech and music. Understanding the implications of long-term analyses, such as spectral and connectivity analyses over extended periods of around 10 minutes, becomes challenging when they do not account for the variable, sometimes quasi-periodical or even non-periodical, elements present in natural speech and music. When contrasting this study with prior research and highlighting its advantages, a more balanced perspective would have been beneficial in the manuscript.

Regarding ecological validity, we respectfully hold a differing perspective from the reviewer. In our view, a one-second music stimulus lacks ecological validity, as real-world music always extends much beyond such a brief duration. While we acknowledge the trade-off in selecting longer stimuli, limiting the diversity of musical styles, we maintain that only long stimuli afford participants an authentic musical listening experience. Conversely, shorter stimuli may lead participants to merely "skip through" musical excerpts rather than engage in genuine listening.

Regarding the critique that we "did not distinguish between neural responses to the varied contents or structures within speech and music," we partly concur. Our TRF (temporal response function) analyzes incorporate acoustic content, particularly the acoustic envelope, thereby addressing this concern to some extent. However, it is accurate to note that we did not model non-acoustic features. In acknowledging this limitation, we would like to share an additional thought with the reviewer regarding model comparison for speech and music. Specifically, comparing results from a phonetic (or syntactic) model of speech to a pitch-melodic (or harmonic) model for music is not straightforward, as these models operate on fundamentally different dimensions. In other words, while assuming equivalence between phonemes and pitches may be a reasonable assumption, it in essence relies on a somewhat arbitrary choice. Consequently, comparing and interpreting neuronal population coding for one or the other model remains problematic. In summary, because the models for speech and music are different (except for acoustic models), direct comparison is challenging, although still commendable and of interest.

Finally, we did take into account the reviewer’s remark and did our best to give a more balanced perspective of our approach and previous studies in the discussion.

“While listening to natural speech and music rests on cognitively relevant neural processes, our analytical approach, extending over a rather long period of time, does not allow to directly isolate specific brain operations. Computational models -which can be as diverse as acoustic (Chi et al., 2005), cognitive (Giordano et al., 2021), information-theoretic (Di Liberto et al., 2020), or self-supervised neural network (Donhauser & Baillet, 2019 ; Millet et al., 2022) models- are hence necessary to further our understanding of the type of computations performed by our reported frequency-specific distributed networks. Moreover, incorporating models accounting for musical and linguistic structure can help us avoid misattributing differences between speech and music driven by unmatched sensitivity factors (e.g., arousal, emotion, or attention) as inherent speech or music selectivity (Mas-Herrero et al., 2013; Nantais & Schellenberg, 1999).”

(2) In contrast to previous studies that employed short stimulus segments along with various control stimuli to ensure that observed selectivity for speech or music was not merely due to low-level acoustic properties, this study used longer, ecological stimuli. However, the control stimuli used in this study, such as tone or syllable sequences, do not align with the low-level acoustic properties of the speech and music stimuli. This mismatch raises concerns that the differences or selectivity between speech and music observed in this study might be attributable to these basic acoustic characteristics rather than to more complex processing factors specific to speech or music.

We acknowledge the reviewer's concern. Indeed, speech and music differ on various levels, including acoustic and cognitive aspects, and our analyzes do not explicitly distinguish them. The aim of this study was to provide an overview of the similarities and differences between natural speech and music processing, in ecological context. Future work is needed to explore further the different hierarchical levels or networks composing such listening experiences. Of note, however, we report whole-brain results with high spatial resolution (thanks to iEEG recordings), enabling the distinction between auditory, superior temporal gyrus (STG), and higher-level responses. Our findings clearly highlight that both auditory and higher-level regions predominantly exhibit shared responses, challenging the interpretation that our results can be attributed solely to differences in 'basic acoustic characteristics'.

We have now more clearly pointed out this reasoning in the results section:

“The spatial distribution of the spectrally-resolved responses corresponds to the network typically involved in speech and music perception. This network encompasses both ventral and dorsal auditory pathways, extending well beyond the auditory cortex and, hence, beyond auditory processing that may result from differences in the acoustic properties of our baseline and experimental stimuli.“

(3) The concept of selectivity - shared, preferred, and domain-selective - increases the risks of potentially overgeneralized interpretations and theoretical inaccuracies. The authors' categorization of neural sites/regions as shared, preferred, or domain-selective regarding speech and music processing essentially resembles a traditional ANOVA test with post hoc analysis. While this categorization gives meaningful context to the results, the mere presence of significant differences among control stimuli, a segment of speech, and a piece of music does not necessarily imply that a region is specifically selective to a type of stimulus like speech. The manuscript's narrative might lead to an overgeneralized interpretation that their findings apply broadly to speech or music. However, identifying differences in neural responses to a few sets of specific stimuli in one brain region does not robustly support such a generalization. This is because speech and music are inherently diverse, and specificity often relates more to the underlying functions than to observed neural responses to a limited number of examples of a stimulus type. See the next point.

Exactly! Here, we present a precise operational definition of these terms, implemented with clear and rigorous statistical methods. It is important to note that in many cognitive neuroscience studies, the term "selective" is often used without a clear definition. By establishing operational definitions, we identified three distinct categories based on statistical testing of differences from baseline and between conditions. This approach provides a framework for more accurate interpretation of experimental findings, as now better outlined in the introduction:

“Finally, we suggest that terms should be operationally defined based on statistical tests, which results in a clear distinction between shared, selective, and preferred activity. That is, be A and B two investigated cognitive functions, “shared” would be a neural population that (compared to a baseline) significantly and equally contributes to the processing of both A and B; “selective” would be a neural population that exclusively contributes to the processing of A or B (e.g. significant for A but not B); and “preferred” would be a neural population that significantly contributes to the processing of both A and B, but more prominently for A or B (Figure 1A).”

Regarding the risk of over-generalization, we want to clarify that our manuscript does not claim that a specific region or frequency band is selective to speech or music. As indeed we focus on testing excerpts of speech and music, we employ the reverse logical reasoning: "if 10 minutes of instrumental music activates a region traditionally associated with speech selectivity, we can conclude that this region is NOT speech-selective." Our conclusions revolve around the absence of selectivity rather than the presence of selective areas or frequency bands. In essence, "one counterexample is enough to disprove a theory." We now further elaborated on this point in the discussion section:

“In this context, in the current study we did not observe a single anatomical region for which speech-selectivity was present, in any of our analyzes. In other words, 10 minutes of instrumental music was enough to activate cortical regions classically labeled as speech (or language) -selective. On the contrary, we report spatially distributed and frequency-specific patterns of shared, preferred, or selective neural responses and connectivity fingerprints. This indicates that domain-selective brain regions should be considered as a set of functionally homogeneous but spatially distributed voxels, instead of anatomical landmarks.”

(4) The authors' approach, akin to mapping a 'receptive field' by correlating stimulus properties with neural responses to ascertain functional selectivity for speech and music, presents issues. For instance, in the cochlea, different stimuli activate different parts of the basilar membrane due to the distinct spectral contents of speech and music, with each part being selective to certain frequencies. However, this phenomenon reflects the frequency selectivity of the basilar membrane - an important function, not an inherent selectivity for speech or music. Similarly, if cortical regions exhibit heightened responses to one type of stimulus over another, it doesn't automatically imply selectivity or preference for that stimulus. The explanation could lie in functional aspects, such as a region's sensitivity to temporal units of a specific duration, be it music, speech, or even movie segments, and its role in chunking such units (e.g., around 500 ms), which might be more prevalent in music than in speech, or vice versa in the current study. This study does not delve into the functional mechanisms of how speech and music are processed across different musical or linguistic hierarchical levels but merely demonstrates differences in neural responses to various stimuli over a 10-minute span.

We completely agree with the last statement, as our primary goal was not to investigate the functional mechanisms underlying speech and music processing. However, the finding of a substantial portion of the cortical network as being shared between the two domains constrains our understanding of the underlying common operations. Regarding the initial part of the comment, we would like to clarify that in the framework we propose, if cortical regions show heightened responses to one type of stimulus over another, this falls into the ‘preferred’ category. The ‘selective’ (exclusive) category, on the other hand, would require that the region be unresponsive to one of the two stimuli.

**Reviewer #2 (Public Review):**
Summary:The study investigates whether speech and music processing involve specific or shared brain networks. Using intracranial EEG recordings from 18 epilepsy patients, it examines neural responses to speech and music. The authors found that most neural activity is shared between speech and music processing, without specific regional brain selectivity. Furthermore, domain-selective responses to speech or music are limited to frequency-specific coherent oscillations. The findings challenge the notion of anatomically distinct regions for different cognitive functions in the auditory process.Strengths:(1) This study uses a relatively large corpus of intracranial EEG data, which provides high spatiotemporal resolution neural recordings, allowing for more precise and dynamic analysis of brain responses. The use of continuous speech and music enhances ecological validity compared to artificial or segmented stimuli.(2) This study uses multiple frequency bands in addition to just high-frequency activity (HFA), which has been the focus of many existing studies in the literature. This allows for a more comprehensive analysis of neural processing across the entire spectrum. The heterogeneity across different frequency bands also indicates that different frequency components of the neural activity may reflect different underlying neural computations.(3) This study also adds empirical evidence towards distributed representation versus domain-specificity. It challenges the traditional view of highly specialized, anatomically distinct regions for different cognitive functions. Instead, the study suggests a more integrated and overlapping neural network for processing complex stimuli like speech and music.Weaknesses:While this study is overall convincing, there are still some weaknesses in the methods and analyses that limit the implication of the work.The study's main approach, focusing primarily on the grand comparison of response amplitudes between speech and music, may overlook intricate details in neural coding. Speech and music are not entirely orthogonal with each other at different levels of analysis: at the high-level abstraction, these are two different categories of cognitive processes; at the low-level acoustics, they overlap a lot; at intermediate levels, they may also share similar features. The selected musical stimuli, incorporating both vocals and multiple instrumental sounds, raise questions about the specificity of neural activation. For instance, it's unclear if the vocal elements in music and speech engage identical neural circuits. Additionally, the study doesn't adequately address whether purely melodic elements in music correlate with intonations in speech at a neural level. A more granular analysis, dissecting stimuli into distinct features like pitch, phonetics, timbre, and linguistic elements, could unveil more nuanced shared, and unique neural processes between speech and music. Prior research indicates potential overlap in neural coding for certain intermediate features in speech and music (Sankaran et al. 2023), suggesting that a simple averaged response comparison might not fully capture the complexity of neural encoding. Further delineation of phonetic, melodic, linguistic, and other coding, along with an analysis of how different informational aspects (phonetic, linguistic, melodic, etc) are represented in shared neural activities, could enhance our understanding of these processes and strengthen the study's conclusions.

We appreciate the reviewer's acknowledgment that delving into the intricate details of neural coding of speech and music was beyond the scope of this work. To address some of the more precise issues raised, we have clarified in the manuscript that our musical stimuli do not contain vocals and are purely instrumental. We apologize if this was not clear initially.

“In the main experimental session, patients passively listened to ~10 minutes of storytelling (Gripari, 2004); 577 secs, La sorcière de la rue Mouffetard, (Gripari, 2004) and ~10 minutes of instrumental music (580 secs, Reflejos del Sur, Oneness, 2006) separated by 3 minutes of rest.”

Furthermore, we now acknowledge the importance of modeling melodic, phonetic, or linguistic features in the discussion, and we have referenced the work of Sankaran et al. (2024) and McCarty et al. (2023) in this regard. However, we would like to share an additional thought with the reviewer regarding model comparison for speech and music. Specifically, comparing results from a phonetic (or syntactic) model of speech to a pitch-melodic (or harmonic) model for music is not straightforward, as these models operate on fundamentally different dimensions. In other words, while assuming equivalence between phonemes and pitches may be a reasonable assumption, it in essence relies on a somewhat arbitrary choice. Consequently, comparing and interpreting neuronal population coding for one or the other model remains problematic. In summary, because the models for speech and music are different (except for acoustic models), direct comparison is challenging, although still commendable and of interest.

“These selective responses, not visible in primary cortical regions, seem independent of both low-level acoustic features and higher-order linguistic meaning (Norman-Haignere et al., 2015), and could subtend intermediate representations (Giordano et al., 2023) such as domain-dependent predictions (McCarty et al., 2023; Sankaran et al., 2023).”

References:

McCarty, M. J., Murphy, E., Scherschligt, X., Woolnough, O., Morse, C. W., Snyder, K., Mahon, B. Z., & Tandon, N. (2023). Intraoperative cortical localization of music and language reveals signatures of structural complexity in posterior temporal cortex. *iScience, 26(7)*, 107223.

Sankaran, N., Leonard, M. K., Theunissen, F., & Chang, E. F. (2023). Encoding of melody in the human auditory cortex. *bioRxiv*. https://doi.org/10.1101/2023.10.17.562771

The paper's emphasis on shared and overlapping neural activity, as observed through sEEG electrodes, provides valuable insights. It is probably true that domain-specificity for speech and music does not exist at such a macro scale. However, it's important to consider that each electrode records from a large neuronal population, encompassing thousands of neurons. This broad recording scope might mask more granular, non-overlapping feature representations at the single neuron level. Thus, while the study suggests shared neural underpinnings for speech and music perception at a macroscopic level, it cannot definitively rule out the possibility of distinct, non-overlapping neural representations at the microscale of local neuronal circuits for features that are distinctly associated with speech and music. This distinction is crucial for fully understanding the neural mechanisms underlying speech and music perception that merit future endeavors with more advanced large-scale neuronal recordings.

We appreciate the reviewer's concern, but we do not view this as a weakness for our study's purpose. Every method inherently has limitations, and intracranial recordings currently offer the best possible spatial specificity and temporal resolution for studying the human brain. Studying cell assemblies thoroughly in humans is ethically challenging, and examining speech and music in non-human primates or rats raises questions about cross-species analogy. Therefore, despite its limitations, we believe intracranial recording remains the best option for addressing these questions in humans.

Regarding the granularity of neural representation, while understanding how computations occur in the central nervous system is crucial, we question whether the single neuron scale provides the most informative insights. The single neuron approach seem more versatile (e.g., in term of cell type or layer affiliation) than the local circuitry they contribute to, which appears to be the brain's building blocks (e.g., like the laminar organization; see Mendoza-Halliday et al.,2024). Additionally, the population dynamics of these functional modules appear crucial for cognition and behavior (Safaie et al. 2023; Buzsáki and Vöröslakos, 2023). Therefore, we emphasize the need for multi-scale research, as we believe that a variety of approaches will complement each other's weaknesses when taken individually. We clarified this in the introduction:

“This approach rests on the idea that the canonical computations that underlie cognition and behavior are anchored in population dynamics of interacting functional modules (Safaie et al. 2023; Buzsáki and Vöröslakos, 2023) and bound to spectral fingerprints consisting of network- and frequency-specific coherent oscillations (Siegel et al., 2012).”

Importantly, we focus on the macro-scale and conclude that, at the anatomical region level, no speech or music selectivity can be observed during natural stimulation. This is stated in the discussion, as follow:

“In this context, in the current study we did not observe a single anatomical region for which speech-selectivity was present, in any of our analyses. In other words, 10 minutes of instrumental music was enough to activate cortical regions classically labeled as speech (or language) -selective. On the contrary, we report spatially distributed and frequency-specific patterns of shared, preferred, or selective neural responses and connectivity fingerprints. This indicates that domain-selective brain regions should be considered as a set of functionally homogeneous but spatially distributed voxels, instead of anatomical landmarks.”

References :

Mendoza-Halliday, D., Major, A.J., Lee, N. *et al.* A ubiquitous spectrolaminar motif of local field potential power across the primate cortex. *Nat Neurosci* (2024).

Safaie, M., Chang, J.C., Park, J. *et al.* Preserved neural dynamics across animals performing similar behaviour. *Nature*
**623**, 765–771 (2023).

Buzsáki, G., & Vöröslakos, M. (2023). Brain rhythms have come of age. *Neuron*, *111*(7), 922-926.

While classifying electrodes into 3 categories provides valuable insights, it may not fully capture the complexity of the neural response distribution to speech and music. A more nuanced and continuous approach could reveal subtler gradations in neural response, rather than imposing categorical boundaries. This could be done by computing continuous metrics, like unique variances explained by each category, or ratio-based statistics, etc. Incorporating such a continuum could enhance our understanding of the neural representation of speech and music, providing a more detailed and comprehensive picture of cortical processing.

To clarify, the metrics we are investigating (coherence, power, linear correlations) are continuous. Additionally, we conduct a comprehensive statistical analysis of these results. The statistical testing, which includes assessing differences from baseline and between the speech and music conditions using a statistical threshold, yields three categories. Of note, ratio-based statistics (a continuous metric) are provided in Figures S9 and S10 (Figures S8 and S9 in the original version of the manuscript).

**Reviewer #3 (Public Review):**
Summary:Te Rietmolen et al., investigated the selectivity of cortical responses to speech and music stimuli using neurosurgical stereo EEG in humans. The authors address two basic questions: 1. Are speech and music responses localized in the brain or distributed; 2. Are these responses selective and domain-specific or rather domain-general and shared? To investigate this, the study proposes a nomenclature of shared responses (speech and music responses are not significantly different), domain selective (one domain is significant from baseline and the other is not), domain preferred (both are significant from baseline but one is larger than the other and significantly different from each other). The authors employ this framework using neural responses across the spectrum (rather than focusing on high gamma), providing evidence for a low level of selectivity across spectral signatures. To investigate the nature of the underlying representations they use encoding models to predict neural responses (low and high frequency) given a feature space of the stimulus envelope or peak rate (by time delay) and find stronger encoding for both in the low-frequency neural responses. The top encoding electrodes are used as seeds for a pair-wise connectivity (coherence) in order to repeat the shared/selective/preferred analysis across the spectra, suggesting low selectivity. Spectral power and connectivity are also analyzed on the level of the regional patient population to rule out (and depict) any effects driven by a select few patients. Across analyses the authors consistently show a paucity of domain selective responses and when evident these selective responses were not represented across the entire cortical region. The authors argue that speech and music mostly rely on shared neural resources.Strengths:I found this manuscript to be rigorous providing compelling and clear evidence of shared neural signatures for speech and music. The use of intracranial recordings provides an important spatial and temporal resolution that lends itself to the power, connectivity, and encoding analyses. The statistics and methods employed are rigorous and reliable, estimated based on permutation approaches, and cross-validation/regularization was employed and reported properly. The analysis of measures across the entire spectra in both power, coherence, and encoding models provides a comprehensive view of responses that no doubt will benefit the community as an invaluable resource. Analysis of the level of patient population (feasible with their high N) per region also supports the generalizability of the conclusions across a relatively large cohort of patients. Last but not least, I believe the framework of selective, preferred, and shared is a welcome lens through which to investigate cortical function.Weaknesses:I did not find methodological weaknesses in the current version of the manuscript. I do believe that it is important to highlight that the data is limited to passively listening to naturalistic speech and music. The speech and music stimuli are not completely controlled with varying key acoustic features (inherent to the different domains). Overall, I found the differences in stimulus and lack of attentional controls (passive listening) to be minor weaknesses that would not dramatically change the results or conclusions.

Thank you for this positive review of our work. We added these points as limitations and future directions in the discussion section:

“Finally, in adopting here a comparative approach of speech and music – the two main auditory domains of human cognition – we only investigated one type of speech and of music also using a passive listening task. Future work is needed to investigate for instance whether different sentences or melodies activate the same selective frequency-specific distributed networks and to what extent these results are related to the passive listening context compared to a more active and natural context (e.g. conversation).”

**Recommendations for the authors:**

**Reviewer #1 (Recommendations For The Authors):**
(1) The concepts of activation and deactivation within the study's context of selectivity are not straightforward to comprehend. It would be beneficial for the authors to provide more detailed explanations of how these phenomena relate to the selectivity of neural responses to speech and music. Such elaboration would aid readers in better understanding the nuances of how certain brain regions are selectively activated or deactivated in response to different auditory stimuli.

The reviewer is right that the reported results are quite complex to interpret. The concepts of activation and deactivation are generally complex to comprehend as they are in part defined by an approach (e.g., method and/or metric) and the scale of observation (Pfurtscheller et al., 1999). The power (or the magnitude) of time-frequency estimate is by definition a positive value. Deactivation (or desynchronization) is therefore related to the comparison used (e.g., baseline, control, condition). This is further complexified by the scale of the measurement, for instance, when it comes to a simple limb movement, some brain areas in sensory motor cortex are going to be activated, yet this phenomenon is accompanied at a finer scale by some desynchonization of the mu-activity, and such desynchronization is a relative measure (e.g., before/after motor movement). At a broader scale it is not rare to see some form of balance between brain networks, some being ‘inhibited’ to let some others be activated like the default mode network versus sensory-motor networks. In our case, when estimating selective responses, it is the strength of the signal that matters. The type of selectivity is then defined by the sign/direction of the comparison/subtraction. We now provide additional details about the sign of selectivity between domains and frequencies in the Methods and Results section:

Methods:

“In order to explore the full range of possible selective, preferred, or shared responses, we considered both responses greater and smaller than the baseline. Indeed, as neural populations can synchronize or desynchronize in response to sensory stimulation, we estimated these categories separately for significant activations and significant deactivations compared to baseline.”

Results:

“We classified, for each canonical frequency band, each channel into one of the categories mentioned above, i.e. shared, selective, or preferred (Figure 1A), by examining whether speech and/or music differ from baseline and whether they differ from each other. We also considered both activations and deactivations, compared to baseline, as both index a modulation of neural population activity, and have been linked with cognitive processes (Pfurtscheller & Lopes da Silva, 1999; Proix et al., 2022). However, because our aim was not to interpret specific increase or decrease with respect to the baseline, we here simply consider significant deviations from the baseline. In other words, when estimating selectivity, it is the strength of the response that matters, not its direction (activation, deactivation).”

“Both domains displayed a comparable percentage of selective responses across frequency bands (Figure 4, first values of each plot). When considering separately activation (Figure 2) and deactivation (Figure 3) responses, speech and music showed complementary patterns: for low frequencies (<15 Hz) speech selective (and preferred) responses were mostly deactivations and music responses activations compared to baseline, and this pattern reversed for high frequencies (>15 Hz).”

References :

J.P. Lachaux, J. Jung, N. Mainy, J.C. Dreher, O. Bertrand, M. Baciu, L. Minotti, D. Hoffmann, P. Kahane,Silence Is Golden: Transient Neural Deactivation in the Prefrontal Cortex during Attentive Reading, *Cerebral Cortex*, Volume 18, Issue 2, February 2008, Pages 443–450

Pfurtscheller, G., & Da Silva, F. L. (1999). Event-related EEG/MEG synchronization and desynchronization: basic principles. Clinical neurophysiology, 110(11), 1842-1857

(2) The manuscript doesn't easily provide information about the control conditions, yet the conclusion significantly depends on these conditions as a baseline. It would be beneficial if the authors could clarify this information for readers earlier and discuss how their choice of control stimuli influences their conclusions.

We added information in the Results section about the baseline conditions:

“[...] with respect to two baseline conditions, in which patients passively listened to more basic auditory stimuli: one in which patients passively listened to pure tones (each 30 ms in duration), the other in which patients passively listened to isolated syllables (/ba/ or /pa/, see Methods).”

Of note, while the choice of different ‘basic auditory stimuli’ as baseline can change the reported results in regions involved in low-level acoustical analyzes (auditory cortex), it will have no impact on the results observed in higher-level regions, which predominantly also exhibit shared responses. We have now more clearly pointed out this reasoning in the results section:

“The spatial distribution of the spectrally-resolved responses corresponds to the network typically involved in speech and music perception. This network encompasses both ventral and dorsal auditory pathways, extending well beyond the auditory cortex and, hence, beyond auditory processing that may result from differences in the acoustic properties of our baseline and experimental stimuli.“

(3) The spectral analyses section doesn't clearly explain how the authors performed multiwise correction. The authors' selectivity categorization appears similar to ANOVAs with posthoc tests, implying the need for certain corrections in the p values or categorization. Could the authors clarify this aspect?

We apologize that this was not in the original version of the manuscript. In the spectral analyzes, the selectivity categorization depended on both (1) the difference effects between the domains and the baseline, and (2) the difference effect between domains. Channels were marked as selective when there was (1) a significant difference between domains and (2) only one domain significantly differed from the baseline. All difference effects were estimated using the paired sample permutation tests based on the t-statistic from the mne-python library (Gramfort et al., 2014) with 1000 permutations and the build-in *tmax* method to correct for the multiple comparisons over channels (Nichols & Holmes, 2002; Groppe et al. 2011). We have now more clearly explained how we controlled family-wise error in the Methods section:

“For each frequency band and channel, the statistical difference between conditions was estimated with paired sample permutation tests based on the t-statistic from the mne-python library (Gramfort et al., 2014) with 1000 permutations and the tmax method to control the family-wise error rate (Nichols and Holmes 2002; Groppe et al. 2011). In tmax permutation testing, the null distribution is estimated by, for each channel (i.e. each comparison), swapping the condition labels (speech vs music or speech/music vs baseline) between epochs. After each permutation, the most extreme t-scores over channels (tmax) are selected for the null distribution. Finally, the t-scores of the observed data are computed and compared to the simulated tmax distribution, similar as in parametric hypothesis testing. Because with an increased number of comparisons, the chance of obtaining a large tmax (i.e. false discovery) also increases, the test automatically becomes more conservative when making more comparisons, as such correcting for the multiple comparison between channels.”

References :

Gramfort, A., Luessi, M., Larson, E., Engemann, D. A., Strohmeier, D., Brodbeck, C., Parkkonen, L., & Hämäläinen, M. S. (2014). MNE software for processing MEG and EEG data. NeuroImage, 86, 446–460.

Groppe, D. M., Bickel, S., Dykstra, A. R., Wang, X., Mégevand, P., Mercier, M. R., Lado, F. A., Mehta, A. D., & Honey, C. J. (2017). iELVis: An open source MATLAB toolbox for localizing and visualizing human intracranial electrode data. Journal of Neuroscience Methods, 281, 40–48.

Nichols, T. E., & Holmes, A. P. (2002). Nonparametric permutation tests for functional neuroimaging: a primer with examples. Human Brain Mapping, 15(1), 1–25.

**Reviewer #2 (Recommendations For The Authors):**
Other suggestions:(1) The authors need to provide more details on how the sEEG electrodes were localized and selected. Are all electrodes included or only the ones located in the gray matter? If all electrodes were used, how to localize and label the ones that are outside of gray matter? In Figures 1C & 1D it seems that a lot of the electrodes were located in depth locations, how were the anatomical labels assigned for these electrodes

We apologize that this was not clear in the original version of the manuscript. Our electrode localization procedure was based on several steps described in detail in Mercier et al., 2022. Once electrodes were localized in a post-implant CT-scan and the coordinates projected onto the pre-implant MRI, we were able to obtain the necessary information regarding brain tissues and anatomical region. That is, first, the segmentation of the pre-impant MRI with SPM12 provided both the tissue probability maps (i.e. gray, white, and cerebrospinal fluid (csf) probabilities) and the indexed-binary representations (i.e., either gray, white, csf, bone, or soft tissues) that allowed us to dismiss electrodes outside of the brain and select those in the gray matter. Second, the individual's brain was co-registered to a template brain, which allowed us to back project atlas parcels onto individual’s brain and assign anatomical labels to each electrode. The result of this procedure allowed us to group channels by anatomical parcels as defined by the Brainnetome atlas (Figure 1D), which informed the analyses presented in section Population Prevalence (Methods, Figures 4, 9-10, S4-5). Because this study relies on stereotactic EEG, and not Electro-Cortico-Graphy, recording sites include both gyri and sulci, while depth structures were not retained.

We have now updated the “General preprocessing related to electrodes localisation” section in the Methods. The relevant part now states:

“To precisely localize the channels, a procedure similar to the one used in the iELVis toolbox and in the fieldtrip toolbox was applied (Groppe et al., 2017; Stolk et al., 2018). First, we manually identified the location of each channel centroid on the post-implant CT scan using the Gardel software (Medina Villalon et al., 2018). Second, we performed volumetric segmentation and cortical reconstruction on the pre-implant MRI with the Freesurfer image analysis suite (documented and freely available for download online http://surfer.nmr.mgh.harvard.edu/). This segmentation of the pre-implant MRI with SPM12 provides us with both the tissue probability maps (i.e. gray, white, and cerebrospinal fluid (CSF) probabilities) and the indexed-binary representations (i.e., either gray, white, CSF, bone, or soft tissues). This information allowed us to reject electrodes not located in the brain. Third, the post-implant CT scan was coregistered to the pre-implant MRI via a rigid affine transformation and the pre-implant MRI was registered to MNI152 space, via a linear and a non-linear transformation from SPM12 methods (Penny et al., 2011), through the FieldTrip toolbox (Oostenveld et al., 2011). Fourth, applying the corresponding transformations, we mapped channel locations to the pre-implant MRI brain that was labeled using the volume-based Human Brainnetome Atlas (Fan et al., 2016).”

Reference:

Mercier, M. R., Dubarry, A.-S., Tadel, F., Avanzini, P., Axmacher, N., Cellier, D., Vecchio, M. D., Hamilton, L. S., Hermes, D., Kahana, M. J., Knight, R. T., Llorens, A., Megevand, P., Melloni, L., Miller, K. J., Piai, V., Puce, A., Ramsey, N. F., Schwiedrzik, C. M., … Oostenveld, R. (2022). Advances in human intracranial electroencephalography research, guidelines and good practices. NeuroImage, 260, 119438.

(2) From Figures 5 and 6 (and also S4, S5), is it true that aside from the shared response, lower frequency bands show more music selectivity (blue dots), while higher frequency bands show more speech selectivity (red dots)? I am curious how the authors interpret this.

The reviewer is right in noticing the asymmetric selective response to music and speech in lower and higher frequency bands. However, while this effect is apparent in the analyzes wherein we inspected stronger synchronization (activation) compared to baseline (Figures 2 and S1), the pattern appears to reverse when examining deactivation compared to baseline (Figures 3 and S2). In other words, there seems to be an overall stronger deactivation for speech in the lower frequency bands and a relatively stronger deactivation for music in the higher frequency bands.

We now provide additional details about the sign of selectivity between domains and frequencies in the Results section:

“Both domains displayed a comparable percentage of selective responses across frequency bands (Figure 4, first values of each plot). When considering separately activation (Figure 2) and deactivation (Figure 3) responses, speech and music showed complementary patterns: for low frequencies (<15 Hz) speech selective (and preferred) responses were mostly deactivations and music responses activations compared to baseline, and this pattern reversed for high frequencies (>15 Hz).”

Note, however, that this pattern of results depends on only a select number of patients, i.e. when ignoring regional selective responses that are driven by as few as 2 to 4 patients, the pattern disappears (Figures 5-6). More precisely, ignoring regions explored by a small number of patients almost completely clears the selective responses for both speech and music. For this reason, we do not feel confident interpreting the possible asymmetry in low vs high frequency bands differently encoding (activation or deactivation) speech and music.

Minor:(1) P9 L234: Why only consider whether these channels were unresponsive to the other domain in the other frequency bands? What about the responsiveness to the target domain?

We thank the reviewer for their interesting suggestion. The primary objective of the cross-frequency analyzes was to determine whether domain-selective channels for a given frequency band remain unresponsive (i.e. exclusive) to the other domain across frequency bands, or whether the observed selectivity is confined to specific frequency ranges (i.e.frequency-specific). In other words, does a given channel exclusively respond to one domain and never—in whichever frequency band—to the other domain? The idea behind this question is that, for a channel to be selectively involved in the encoding of one domain, it does not necessarily need to be sensitive to all timescales underlying that domain as long as it remains unresponsive to any timescale in the other domain. However, if the channel is sensitive to information that unfolds slowly in one domain and faster in the other domain, then the channel is no longer globally domain selective, but the selectivity is frequency-specific to each domain.

The proposed analyzes answer a slightly different, albeit also meaningful, question: how many frequencies (or frequency bands) do selective responses span? From the results presented below, the reviewer can appreciate the overall steep decline in selective response beyond the single frequency band with only few channels remaining selectively responsive across maximally four frequency bands. That is, selective responses globally span one frequency band.

**Author response image 1. sa4fig1:** Cross-frequency channel selective responses. The top figure shows the results for the spectral analyzes (baselined against the tones condition, including both activation and deactivation). The bottom figure shows the results for the connectivity analyzes. For each plot, the first (leftmost) value corresponds to the percentage (%) of channels displaying a selective response in a specific frequency band. In the next value, we remove the channels that no longer respond selectively to the target domain for the following frequency band. The black dots at the bottom of the graph indicate which frequency bands were successively included in the analysis.

(2) P21 L623: "Population prevalence." The subsection title should be in bold.

Done.

**Reviewer #3 (Recommendations For The Authors):**
The authors chose to use pure tone and syllables as baseline, I wonder if they also tried the rest period between tasks and if they could comment on how it differed and why they chose pure tones, (above and beyond a more active auditory baseline).

This is an interesting suggestion. The reason for not using the baseline between speech and music listening (or right after) is that it will be strongly influenced by the previous stimulus. Indeed, after listening to the story it is likely that patients keep thinking about the story for a while. Similarly after listening to some music, the music remains in “our head” for some time.

This is why we did not use rest but other auditory stimulation paradigms. Concerning the choice of pure tones and syllables, these happen to be used for clinical purposes to assess functioning of auditory regions. They also corresponded to a passive listening paradigm, simply with more basic auditory stimuli. We clarified this in the Results section:

“[...] with respect to two baseline conditions, in which patients passively listened to more basic auditory stimuli: one in which patients passively listened to pure tones (each 30 ms in duration), the other in which patients passively listened to isolated syllables (/ba/ or /pa/, see Methods).”

Discussion - you might want to address phase information in contrast to power. Your encoding models map onto low-frequency (bandpassed) activity which includes power and phase. However, the high-frequency model includes only power. The model comparison is not completely fair and may drive part of the effects in Figure 7a. I would recommend discussing this, or alternatively ruling out the effect with modeling power separately for the low frequency.

We thank the reviewer for their recommendation. First, we would like to emphasize that the chosen signal extraction techniques that we used are those most frequently reported in previous papers (e.g. Ding et al., 2012; Di Liberto et al., 2015; Mesgarani and Chang, 2012).

Low-frequency (LF) phase and high-frequency (HFa) amplitude are also known to track acoustic rhythms in the speech signal in a joint manner (Zion-Golumbic et al., 2013; Ding et al., 2016). This is possibly due to the fact that HFa amplitude and LF phase dynamics have a somewhat similar temporal structure (see Lakatos et al., 2005 ; Canolty and Knight, 2010).

Still, the reviewer is correct in pointing out the somewhat unfair model comparison and we appreciate the suggestion to rule out a potential confound. We now report in Supplementary Figure S8, a model comparison for LF amplitude vs. HFa amplitude to complement the findings displayed in Figure 7A. Overall, the reviewer can appreciate that using LF amplitude or phase does not change the results: LF (amplitude or phase) always better captures acoustic features than HFa amplitude.

**Author response image 2. sa4fig2:** TRF model comparison of low-frequency (LF) amplitude and high-frequency (HFa) amplitude. Models were investigated to quantify the encoding of the instantaneous envelope and the discrete acoustic onset edges (peakRate) by either the low frequency (LF) amplitude or the high frequency (HFa) amplitude. The ‘peakRate & LF amplitude’ model significantly captures the largest proportion of channels, and is, therefore, considered the winning model. Same conventions as in Figure 7A.

References:

Canolty, R. T., & Knight, R. T. (2010). The functional role of cross-frequency coupling. Trends in Cognitive Sciences, 14(11), 506–515.

Di Liberto, G. M., O’sullivan, J. A., & Lalor, E. C. (2015). Low-frequency cortical entrainment to speech reflects phoneme-level processing. Current Biology, 25(19), 2457-2465.

Ding, N., & Simon, J. Z. (2012). Emergence of neural encoding of auditory objects while listening to competing speakers. Proceedings of the National Academy of Sciences, 109(29), 11854-11859.

Ding, N., Melloni, L., Zhang, H., Tian, X., & Poeppel, D. (2016). Cortical tracking of hierarchical linguistic structures in connected speech. Nature Neuroscience, 19(1), 158–164.

Golumbic, E. M. Z., Ding, N., Bickel, S., Lakatos, P., Schevon, C. A., McKhann, G. M., ... & Schroeder, C. E. (2013). Mechanisms underlying selective neuronal tracking of attended speech at a “cocktail party”. Neuron, 77(5), 980-991.

Lakatos, P., Shah, A. S., Knuth, K. H., Ulbert, I., Karmos, G., & Schroeder, C. E. (2005). An oscillatory hierarchy controlling neuronal excitability and stimulus processing in the auditory cortex. Journal of Neurophysiology, 94(3), 1904–1911.

Mesgarani, N., & Chang, E. F. (2012). Selective cortical representation of attended speaker in multi-talker speech perception. Nature, 485(7397), 233-236.

Similarly, the Coherence analysis is affected by both power and phase and is not dissociated. i.e. if the authors wished they could repeat the coherence analysis with phase coherence (normalizing by the amplitude). Alternatively, this issue could be addressed in the discussion above

We agree with the Reviewer. We have now better clarified our choice in the Methods section:

“Our rationale to use coherence as functional connectivity metric was three fold. First, coherence analysis considers both magnitude and phase information. While the absence of dissociation can be criticized, signals with higher amplitude and/or SNR lead to better time-frequency estimates (which is not the case with a metric that would focus on phase only and therefore would be more likely to include estimates of various SNR). Second, we choose a metric that allows direct comparison between frequencies. As, at high frequencies phase angle changes more quickly, phase alignment/synchronization is less likely in comparison with lower frequencies. Third, we intend to align to previous work which, for the most part, used the measure of coherence most likely for the reasons explained above.“